# Reassessing Layer Pruning in LLMs: New Insights and Methods

**Yao Lu**[1,2]   **Hao Cheng**[3]   **Yujie Fang**[1]   **Zeyu Wang**[1,2]
**Jiaheng Wei**[4]   **Dongwei Xu**[1,2]   **Qi Xuan**[1,2*]   **Zhaowei Zhu**[5]
[1]Institute of Cyberspace Security, Zhejiang University of Technology
[2]Binjiang Institute of Artificial Intelligence, Zhejiang University of Technology
[3]Hong Kong Baptist University
[4]Hong Kong University of Science and Technology (Guangzhou)   [5]D5 Data
`yaolu.zjut@gmail.com, xuanqi@zjut.edu.cn, zzw@d5data.ai`

## Abstract

Although large language models (LLMs) have achieved remarkable success across various domains, their considerable scale necessitates substantial computational resources, posing significant challenges for deployment in resource-constrained environments. Layer pruning, as a simple yet effective compression method, removes layers of a model directly, reducing computational overhead. However, what are the best practices for layer pruning in LLMs? Are sophisticated layer selection metrics truly effective? Does the LoRA (Low-Rank Approximation) family, widely regarded as a leading method for pruned model fine-tuning, truly meet expectations when applied to post-pruning fine-tuning? To answer these questions, we dedicate thousands of GPU hours to benchmarking layer pruning in LLMs and gaining insights across multiple dimensions. Our results demonstrate that a simple approach, i.e., pruning the final layers followed by fine-tuning the `lm_head` and the remaining last three layers, yields remarkably strong performance. These pruning strategies are further supported by theoretical analyses based on the gradient flow. Following this guide, our method surpasses existing state-of-the-art pruning methods by 5.62%–17.27% on Llama-3.1-8B-It, by 2.36%–19.45% on Llama-3-8B and by 4.34%–9.59% on Llama-3-70B. The code is available at `https://github.com/yaolu-zjut/Navigation_LLM_layer_pruning`.

## 1 Introduction

In recent years, large language models (LLMs) have achieved unprecedented success in many fields, such as text generation (Achiam et al., 2023; Touvron et al., 2023; Dai et al., 2025), semantic analysis (Zeng et al., 2025a;b; Chu et al., 2025) and machine translation (Zhang et al., 2023; Wang et al., 2023). However, these achievements come with massive resource consumption, posing significant challenges for deployment on resource-constrained devices and scenarios (Ke et al., 2024; 2025). To address these challenges, numerous techniques have been developed to create more efficient LLMs, including pruning (Ma et al., 2023; Sun et al., 2023; Lu et al., 2024; Li et al., 2025a; Lu et al., 2025), knowledge distillation (Xu et al., 2024; Huang et al., 2022; Yang et al., 2024b; Wu et al., 2025), quantization (Zhou et al., 2025; Yu et al., 2025), low-rank factorization (Saha et al., 2023; Zhao et al., 2024a), and system-level inference acceleration (Shah et al., 2024; Lee et al., 2024).

Among these methods, pruning has emerged as a promising solution to mitigate the resource demands of LLMs. By selectively removing redundant patterns—such as parameters (Sun et al., 2023), attention heads (Ma et al., 2023) and layers (Men et al., 2024)—pruning aims to slim down the model while maintaining its original performance as much as possible. Among different types of pruning, layer pruning (Kim et al., 2024; Siddiqui et al., 2024) has garnered particular interest due to its direct impact on pruning the model's depth, thereby decreasing both computational complexity and memory usage. Additionally, thanks to the nice structure of the existing LLMs such as Llama (Dubey

---

*Corresponding author: Qi Xuan and Zhaowei Zhu.

et al., 2024), whose transformer blocks have the exactly same dimension of input and output, layer pruning becomes a straightforward and simple solution. Therefore, in this paper, we focus on layer pruning. Unlike existing studies (Men et al., 2024; Yang et al., 2024c; Chen et al., 2024; Zhong et al., 2024; Liu et al., 2024b) that aim to propose various sophisticated pruning methods, we take a step back and focus on the following questions:

**Q1.** *Layer Selection:* Are sophisticated metrics essential for identifying redundant layers?

**Q2.** *Fine-Tuning:* Is the LoRA family the best choice for post-pruning fine-tuning?

To address the aforementioned questions, we conduct extensive benchmarking on layer pruning, dedicating thousands of GPU hours to systematic experiments. Our study spans **7** layer selection metrics, **3** state-of-the-art open-source LLMs, **6** fine-tuning methods and **8** common datasets. From these efforts, we have distilled a practical set of key insights for LLM layer pruning:

1). ***Reverse-order pruning is simple yet effective***, i.e., simply pruning the last several layers performs better than many complex pruning metrics (Kim et al., 2024; Men et al., 2024).

2). ***LoRA performs worse than expected***, i.e., LoRA, the most commonly used fine-tuning method in existing pruning approaches (Sun et al., 2023; Ma et al., 2023; Kim et al., 2024; Men et al., 2024), is **not** the best choice for post-pruning performance recovery. In contrast, freezing the other layers and fine-tuning only the last few remaining layers and *lm_head*, also known as *partial-layer fine-tuning*, can achieve higher accuracy while reducing the training time.

Subsequently, we present an analytical framework based on the gradient flow that demonstrates how gradient weakening in Pre-Layer Normalization (Pre-LN) Transformers reduces the contributions of deep layers. This framework theoretically explains why pruning the final layers and fine-tuning the last few layers of the pruned model are effective. Then, we apply these insights and practices to develop **Llama-3.1-6.3B-It-Alpaca** and **Llama-3-6.3B-Alpaca**. Our method surpasses existing state-of-the-art pruning methods by $5.62\%$–$17.27\%$ on Llama-3.1-8B-It and by $2.36\%$–$19.45\%$ on Llama-3-8B. To validate the scalability of our method, we further extend our evaluation to the Llama-3-70B model. Extensive experiments demonstrate that our method achieves stronger generalization and consistent gains over previous pruning techniques. Finally, we hope our work will help guide future efforts in LLM layer pruning and inform best practices for deploying LLMs in real-world applications. In a nutshell, we make the following contributions:

- *Novel Best Practices:* Through detailed and extensive experiments, we identify reverse-order as a simple and effective layer selection metric and find that partial-layer fine-tuning outperforms LoRA-based techniques.

- *Theoretical Insights:* We present an analytical framework based on the gradient flow, demonstrating how gradient weakening in Pre-LN Transformers reduces the contributions of deep layers, thereby explaining the two key practices identified above.

- *State-of-the-Art Performance:* We carry out extensive experiments on mid-scale (Llama-3.1-8B-It and Llama-3-8B) and large-scale (Llama-3-70B) models. The experimental results consistently demonstrate that our method outperforms existing state-of-the-art pruning methods.

## 2 RELATED WORK

**LLM Layer Pruning.** LLM layer pruning is a technique used to reduce the number of layers in LLMs, aiming to lower computational costs without significantly degrading performance. Specifically, it evaluates the contribution of each layer to the model's overall performance, using criteria such as gradients, activation values, parameter weights, or the layer's influence on the loss function. Layers that contribute the least are then pruned to reduce complexity. For example, LaCo (Yang et al., 2024c) achieves rapid model size reduction by folding subsequent layers into the previous layer, effectively preserving the model structure. Similarly, MKA (Liu et al., 2024b) uses manifold learning and the Normalized Pairwise Information Bottleneck measure (Tishby et al., 2000) to identify the most similar layers for merging. ShortGPT (Men et al., 2024) uses Block Influence (BI) to

measure the importance of each layer in LLMs and remove layers with low BI scores. Kim et al. (2024) utilize Magnitude, Taylor and Perplexity (PPL) to evaluate the significance of each layer. Recently, Song et al. (2025) find that deeper layers are crucial for reasoning and that performance can be recovered through distillation.

**Normalization in Language Models.** Layer Normalization (Lei Ba et al., 2016) plays a crucial role in Transformers (Vaswani, 2017) by directly estimating the normalization statistics from the summed inputs to the neurons within a hidden layer. The original Transformer (Vaswani, 2017) uses Post-LN, which applies layer normalization after the residual connections. However, subsequent research (Baevski & Auli, 2018; Nguyen & Salazar, 2019; Li et al., 2023b; 2025b) shows that putting normalization before the residual connections (Pre-LN) can significantly improve training stability, especially in large language models (Brown et al., 2020; Chaplot, 2023; Touvron et al., 2023). Xiong et al. (2020) theoretically prove that Post-LN results in larger gradients near the output layer, making the use of warm-up essential to prevent instability. By contrast, Pre-LN scales down gradients with the depth of the model, which ensures more stable gradients during initialization.

## 3 AN EMPIRICAL EXPLORATION

This paper aims to contribute to the community the best practice of layer pruning such that practitioners can prune an LLM to an affordable size and desired performance with minimal exploration effort. Specifically, we explore which metric is most effective for identifying unimportant layers and investigate which fine-tuning method most effectively restores model performance after pruning, helping researchers make informed choices.

### 3.1 LAYER PRUNING METRICS

Layer pruning aims to find a subset of unimportant layers such that the pruned model maintains acceptable performance while reducing the model's complexity. Numerous methods have proposed various metrics to identify and prune unimportant layers. Herein, we include 7 popular metrics: Random, Reverse-order, Magnitude-l1 (Kim et al., 2024), Magnitude-l2 (Kim et al., 2024), Taylor, PPL and BI (Men et al., 2024).

Specifically, for the **random selection** baseline, we randomly select several layers to prune. **Reverse-order** (Men et al., 2024) posits that importance is inversely proportional to the sequence order. It assigns lower importance scores to the deeper layers and prune them. **Magnitude** was first introduced by Li et al. (2016) and subsequently adopted by Kim et al. (2024), which assumes that weights exhibiting smaller magnitudes are deemed less informative. Following Kim et al. (2024), we compute $I^n_{\text{Magnitude}} = \sum_k ||W^n_k||_p$, where $W^n_k$ denotes the weight matrix of operation $k$ within the $n$-th transformer layer. In this paper, we uniformly set $p = \{1, 2\}$. As a result, we term these methods as **Magnitude-l1** and **Magnitude-l2**. For a given calibration dataset $D$, the significance of removing weight parameters is indicated by the change in training loss $\mathcal{L} := |\mathcal{L}(W^n_k, D) - \mathcal{L}(W^n_k = 0, D)| \approx |\frac{\partial \mathcal{L}(D)}{\partial W^n_k} W^n_k|$. Following Ma et al. (2023); Kim et al. (2024), we omit the second-order derivatives in this assessment. Then we define the **Taylor** score of the $n$-th transformer layer as $I^n_{\text{Taylor}} = \sum_k |\frac{\partial \mathcal{L}(D)}{\partial W^n_k} W^n_k|$. Following Kim et al. (2024), we remove a single layer and assess its impact on the perplexity (**PPL**) of the pruned model using the calibration dataset $D$. We then prune those layers that lead to a smaller degradation of the PPL. Men et al. (2024) introduce a metric called Block Influence (**BI**) as an effective indicator of layer importance. Specifically, the BI score of the $i$-th layer can be calculated as follows:

$$\text{BI}_i = 1 - \mathbb{E}_{x,t} \frac{x^T_{i,t} x_{i+1,t}}{\|x_{i,t}\|_2 \|x_{i+1,t}\|_2},$$ (1)

where $x_i$ denotes the input of the $i$-th layer and $x_{i,t}$ is the $t$-th row of $x_i$.

### 3.2 EVALUATION AND DATASETS

To assess the performance of the model, we follow the evaluation of Ma et al. (2023) to perform zero-shot task classification on 8 common sense reasoning datasets using the lm-evaluation-harness (Gao et al., 2023) package: MMLU (Hendrycks et al., 2021), CMMLU (Li et al., 2023a),

| Model | Metric | Benchmarks | | | | | | | | Avg Acc↑ |
|---|---|---|---|---|---|---|---|---|---|---|
| | | PIQA↑ | HellaSwag↑ | OpenbookQA↑ | ARC-e↑ | ARC-c↑ | MMLU↑ | CMMLU↑ | WinoGrande↑ | |
| Vicuna-7B-v1.5 | Random | 0.5223 | 0.2607 | 0.1380 | 0.2614 | 0.2176 | 0.2295 | 0.2500 | 0.4672 | 0.2933 |
| | PPL | **0.7361** | 0.4734 | **0.2760** | **0.6705** | 0.3456 | 0.2943 | 0.2569 | 0.5896 | 0.4553 |
| | Magnitude-l1 | 0.5299 | 0.2586 | 0.1440 | 0.2609 | 0.2253 | 0.2297 | 0.2514 | 0.4893 | 0.2986 |
| | Magnitude-l2 | 0.5256 | 0.2578 | 0.1340 | 0.2622 | 0.2108 | 0.2295 | 0.2515 | 0.4838 | 0.2944 |
| | BI | 0.6910 | 0.3987 | 0.2100 | 0.5829 | 0.2654 | 0.2389 | 0.2513 | 0.5036 | 0.3927 |
| | Taylor | 0.5250 | 0.2581 | 0.1360 | 0.2584 | 0.2048 | 0.2318 | 0.2526 | 0.4972 | 0.2955 |
| | Reverse-order | 0.7171 | **0.5005** | 0.2608 | 0.6221 | **0.3848** | **0.4737** | **0.3417** | **0.6267** | **0.4909** |
| Qwen1.5-7B | Random | 0.5408 | 0.2682 | 0.1240 | 0.2630 | 0.2039 | 0.2366 | 0.2457 | 0.4807 | 0.2954 |
| | PPL | 0.7089 | 0.4195 | 0.2240 | 0.5960 | 0.2944 | 0.2457 | 0.2552 | 0.5185 | 0.4078 |
| | Magnitude-l1 | 0.6578 | 0.3989 | 0.2040 | 0.5244 | 0.2901 | 0.2574 | 0.2541 | 0.5249 | 0.3890 |
| | Magnitude-l2 | 0.5903 | 0.3657 | 0.1640 | 0.4630 | 0.2381 | 0.2502 | 0.2513 | 0.5312 | 0.3567 |
| | BI | **0.7220** | 0.4190 | **0.2440** | **0.5972** | 0.2671 | 0.2456 | 0.2536 | 0.5383 | 0.4190 |
| | Taylor | 0.6970 | 0.4284 | 0.2060 | 0.5160 | 0.3140 | **0.5231** | 0.6079 | **0.6046** | 0.4871 |
| | Reverse-order | 0.6942 | **0.4444** | 0.2280 | 0.5143 | **0.3302** | 0.5101 | **0.7171** | 0.5912 | **0.5037** |
| Llama-3.1-8B-It | Random | 0.5653 | 0.2886 | 0.1400 | 0.3169 | 0.1860 | 0.2275 | 0.2559 | 0.5075 | 0.3110 |
| | PPL | **0.7628** | 0.4931 | 0.2640 | **0.7290** | 0.3805 | 0.3367 | 0.2724 | 0.5793 | 0.4772 |
| | Magnitude-l1 | 0.5408 | 0.2634 | 0.1360 | 0.2845 | 0.2014 | 0.2504 | 0.2503 | 0.4878 | 0.3018 |
| | Magnitude-l2 | 0.5413 | 0.2638 | 0.1340 | 0.2841 | 0.2014 | 0.2498 | 0.2504 | 0.4870 | 0.3015 |
| | BI | 0.7176 | 0.4196 | 0.2020 | 0.6107 | 0.2841 | 0.2417 | 0.2494 | 0.5391 | 0.4080 |
| | Taylor | 0.7138 | **0.4964** | 0.2740 | 0.6848 | **0.4181** | 0.2861 | 0.2504 | **0.7135** | 0.4796 |
| | Reverse-order | 0.7002 | 0.4010 | **0.2940** | 0.6170 | 0.3985 | **0.6342** | **0.5449** | 0.6243 | **0.5268** |

Table 1: Zero-shot performance of the pruned models (25% pruning rate, fine-tuning using LoRA). "Avg Acc" denotes the average accuracy calculated among eight datasets. The best pruning results are marked in **boldface**. The sub-optimal ones are underlined. See Table L for more pruning rates.

PIQA (Bisk et al., 2020), HellaSwag (Zellers et al., 2019), WinoGrande (Sakaguchi et al., 2021), ARC-easy (Clark et al., 2018), ARC-challenge (Clark et al., 2018) and OpenbookQA (Mihaylov et al., 2018). See the Appendix for more details on the dataset description.

## 3.3 ARE SOPHISTICATED METRICS ESSENTIAL FOR IDENTIFYING REDUNDANT LAYERS?

The first question is to find the most "redundant" layers to prune. As discussed in the previous subsection , there are various metrics for layer selection, which can be as straightforward as reverse-order, or as complicated as BI. However, does a complicated metric always contribute to a better performance? Probably not. We find that a simple metric, i.e., reverse-order, is competitive among these metrics. Specifically, we conduct comprehensive experiments on Vicuna-7B-v1.5 (Zheng et al., 2024), Qwen1.5-7B (Yang et al., 2024a) and Llama-3.1-8B-Instruct (Dubey et al., 2024). We uniformly prune 8 layers (25% pruning ratio) for Vicuna-7B-v1.5, Qwen1.5-7B and Llama-3.1-8B-Instruct. Experiments with a 50% pruning ratio are provided in Table L. In the fine-tuning stage, we use LoRA with a rank $d$ of 8 and a batch size of 64, and the AdamW optimizer. The learning rate is set to $1 \times 10^{-5}$ with 100 warming steps.

**Results.** As shown in Table 1, we find that the reverse-order metric delivers stable and superior results across various models under the 25% pruning rate, making it a reliable choice for pruning. On average, it outperforms the second-best PPL metric by 6.04% across three models. The result also holds for the 50% pruning rate, as shown in Table L. We hope our insights can help researchers make informed choices when selecting the most suitable pruning metrics for their specific models.

> **Insight #1:** The reverse-order are simple yet foolproof metrics for pruning, providing stable and reliable results across different models and pruning rates.

## 3.4 IS THE LORA FAMILY THE BEST CHOICE FOR POST-PRUNING FINE-TUNING?

In previous studies (Kim et al., 2024; Men et al., 2024), LoRA is often used to restore the performance of pruned models. This raises a question: Is the LoRA family the best choice for post-pruning fine-tuning? To answer this question, we further use QLoRA (Dettmers et al., 2024) and partial-layer fine-tuning techniques to conduct experiments. We briefly introduce these methods as follows:

**LoRA Fine-tuning.** LoRA is one of the best-performing parameter-efficient fine-tuning paradigm that updates dense model layers using pluggable low-rank matrices (Mao et al., 2024). Specifically,

| Model | Method | Layer | Benchmarks | | | | | | | | Avg Acc↑ |
|---|---|---|---|---|---|---|---|---|---|---|---|
| | | | PIQA↑ | HellaSwag↑ | OpenbookQA↑ | ARC-e↑ | ARC-c↑ | MMLU↑ | CMMLU↑ | WinoGrande↑ | |
| Vicuna-7B-v1.5 | LoRA | - | 0.7171 | 0.5005 | 0.2608 | 0.6221 | 0.3848 | 0.4737 | 0.3417 | 0.6267 | 0.4909 |
| | QLoRA | - | 0.6649 | 0.4057 | 0.2700 | 0.5345 | 0.3439 | 0.4809 | 0.3473 | 0.6014 | 0.4561 |
| | Partial-layer | *lm_head only* | 0.7057 | 0.4865 | 0.2880 | 0.6301 | 0.4010 | 0.4819 | 0.3520 | 0.6156 | 0.4951 |
| | | *lm_head+last layer* | 0.7155 | 0.5054 | 0.2900 | 0.6511 | **0.4113** | 0.4831 | 0.3538 | **0.6283** | 0.5048 |
| | | *lm_head+last two layers* | 0.7214 | 0.5060 | **0.3020** | **0.6532** | 0.4002 | 0.4858 | 0.3530 | 0.6267 | **0.5060** |
| | | *lm_head+last three layers* | **0.7247** | **0.5103** | 0.2960 | 0.6528 | 0.3985 | **0.4870** | **0.3544** | 0.6219 | 0.5057 |
| Qwen1.5-7B | LoRA | - | 0.6942 | 0.4444 | 0.2280 | 0.5143 | 0.3302 | 0.5101 | 0.7171 | 0.5912 | 0.5037 |
| | QLoRA | - | 0.6697 | 0.4028 | 0.2400 | 0.4760 | 0.2969 | 0.4797 | 0.6914 | 0.5825 | 0.4799 |
| | Partial-layer | *lm_head only* | 0.7149 | 0.4735 | 0.2460 | 0.5497 | 0.3524 | 0.5467 | 0.7276 | 0.5967 | 0.5259 |
| | | *lm_head+last layer* | 0.7220 | 0.4850 | 0.2440 | 0.5690 | 0.3549 | 0.5719 | **0.7283** | 0.6275 | 0.5378 |
| | | *lm_head+last two layers* | 0.7214 | 0.4915 | **0.2540** | 0.5783 | 0.3584 | 0.5734 | 0.7275 | **0.6298** | 0.5418 |
| | | *lm_head+last three layers* | **0.7296** | **0.4974** | 0.2520 | **0.5808** | **0.3618** | 0.5795 | 0.7272 | 0.6275 | **0.5445** |
| Llama-3.1-8B-It | LoRA | - | 0.7002 | 0.4010 | 0.2940 | 0.6170 | 0.3985 | 0.6342 | 0.5449 | 0.6243 | 0.5268 |
| | QLoRA | - | 0.6980 | 0.3975 | 0.3000 | 0.6183 | 0.3840 | 0.6032 | 0.5090 | 0.6267 | 0.5171 |
| | Partial-layer | *lm_head only* | 0.7334 | 0.4896 | 0.2860 | 0.7012 | 0.4411 | 0.6122 | 0.5442 | **0.6717** | 0.5599 |
| | | *lm_head+last layer* | 0.7350 | 0.5107 | 0.2940 | 0.7193 | 0.4531 | **0.6630** | 0.5526 | 0.6582 | 0.5732 |
| | | *lm_head+last two layers* | 0.7361 | 0.5204 | **0.3080** | 0.7151 | 0.4633 | 0.6588 | **0.5543** | 0.6567 | 0.5766 |
| | | *lm_head+last three layers* | **0.7383** | **0.5323** | **0.3080** | **0.7260** | **0.4684** | 0.6567 | 0.5515 | 0.6646 | **0.5807** |

Table 2: Zero-shot performance of pruned models using various fine-tuning methods under 25% pruning rate (using reverse-order). The best results are marked in **boldface**, and the sub-optimal ones are underlined.

| | LoRA | QLoRA | *lm_head only* | *lm_head+last layer* | *lm_head+last two layers* | *lm_head+last three layers* |
|---|---|---|---|---|---|---|
| Trainable parameters | 15.73M | 15.73M | 525.34M | 743.45M | 961.56M | 1179.68M |
| GPU memory | 45.83G | 14.26G | 39.82G | 42.12G | 44.41G | 48.02G |
| Training time (2 epoch) | 10440.30s | 17249.01s | 6952.92s | 7296.76s | 7616.83s | 7931.36s |

Table 3: The training cost of fine-tuning the pruned Llama-3.1-8B-Instruct (with 8 layers removed in reverse-order) using different methods on 2 NVIDIA A100 GPUs.

for a pre-trained weight matrix $W_0$, LoRA constrains its update by representing the latter with a low-rank decomposition $W_0 + \Delta W = W_0 + BA$. At the beginning of training, $A$ is initialize with a random Gaussian initialization, while $B$ is initialized to zero. During training, $W_0$ is frozen and does not receive gradient updates, while $A$ and $B$ contain trainable parameters. Then the forward pass can be formalized as:

$$W_0 x + \Delta W x = W_0 x + BA x. \tag{2}$$

**QLoRA Fine-tuning.** QLoRA builds on LoRA by incorporating quantization techniques to further reduce memory usage while maintaining, or even enhancing the performance.

**Partial-layer Fine-tuning.** Compared to LoRA and QLoRA, which inject trainable low-rank factorization matrices into each layer, partial-layer fine-tuning simply freezes the weights of some layers while updating only the specified layers to save computing resources and time (Shen et al., 2021; Ngesthi et al., 2021; Peng & Wang, 2020). Following by the common practice of previous studies (Khan & Fang, 2023), we choose to fine-tune only the later layers that are closer to the output, while keeping the earlier layers, which capture more general features, frozen. Specifically, we use two different fine-tuning strategies: one is to finetune only the model head (*lm_head only*), and the other is to finetune the *lm_head* plus the last layer (*lm_head + last layer*), the last two layers (*lm_head + last two layers*), and the last three layers (*lm_head + last three layers*).

In view of the superiority of the reverse-order metric in Section 3.3, we use it to prune here. For the Vicuna-7B-v1.5, Qwen1.5-7B, and Llama-3.1-8B-Instruct models, we prune 8 layers. Subsequently, we utilize LoRA, QLoRA and partial-layer fine-tuning methods to restore performance. To verify the generalizability of these fine-tuning methods, we provide additional fine-tuning results using the Taylor metric in Table M. For fine-tuning with LoRA and partial-layer methods, we utilize the AdamW optimizer, while for QLoRA, we opt for the paged_adamw_8bit optimizer. All other hyperparameter settings are the same as in the previous subsection.

**Results.** As shown in the Table 2 and Table M, we find that fine-tuning with QLoRA slightly hurts the performance of pruned models compared to LoRA. Excitingly, the effect of partial-layer fine-tuning is *significantly better* than LoRA, providing a **viable new direction** for fine-tuning models after pruning. When considering fine-tuning methods for LLMs, in addition to performance, the training cost is also a significant factor to take into account. Therefore, we compare the training

cost of these fine-tuning methods, including training time, gpu memory and trainable parameters. Specifically, we conduct experiments on 2 NVIDIA A100 GPUs using the pruned Llama-3.1-8B-Instruct model (with 8 layers removed in reverse order). Table 3 shows the comparison among these fine-tuning methods. We find that compared to LoRA, partial-layer fine-tuning involves more trainable parameters but maintains comparable GPU usage and achieves faster training time. Additionally, partial-layer fine-tuning outperforms LoRA in effectiveness. In contrast, although QLoRA consumes less GPU memory, it has much longer training time and yields poorer performance. In summary, we conclude that partial-layer fine-tuning is an effective approach to restoring the performance of pruned models.

> **Insight #2:** Partial-layer fine-tuning can serve as an alternative to LoRA, achieving better performance recovery for pruned models while reducing the training time.

## 4 THEORETICAL ANALYSIS

While our extensive experiments in the previous section demonstrate the surprising effectiveness of pruning the last several layers (reverse-order) and fine-tuning the last few remaining layers of the pruned model, these empirical successes beg a fundamental question: Why do these strategies work? We argue that the answer lies in the gradient propagation landscape shaped by the LayerNorm. In this section, we theoretically analyze these key insights

Notably, Most state-of-the-art LLMs, like GPT, LLaMA, and Mistral, adopt the Pre-Layer Normalization (Pre-LN) Transformer architecture. Our goal is to examine whether this architecture weakens the gradients of deep layers during backpropagation. To investigate this, we formally characterize the gradient flow behavior in Pre-LN Transformers and analyze how the gradient norm evolves as it propagates from deep layers to shallow layers.

Specifically, based on an analytical framework of the gradient flow, we explain two key insights: (1) in Pre-LN models, the deeper layers contribute less to overall performance, and (2) fine-tuning the last layer (after pruning several final layers) can recover performance more effectively than applying LoRA updates on all remaining layers.

Consider an L-layer Transformer with a Pre-LN (Dubey et al., 2024) scheme:

$$x_{l+1} = x_l + F\big(\mathrm{LN}(x_l); \theta_l\big), \tag{3}$$

where LN is a normalization operator (LayerNorm (Lei Ba et al., 2016) or RMSNorm (Zhang & Sennrich, 2019)), $x_l$ is the input of the $l$-th layer, $x_{l+1}$ is the output of the $l$-th layer (a.k.a. the input of the $(l+1)$-th layer), $F(\cdot; \theta_l)$ is the learnable sub-layer. We can calculate the derivatives of Equation (3), as follows:

$$\frac{\partial x_{l+1}}{\partial x_l} = I + \underbrace{\frac{\partial F\big(\mathrm{LN}(x_l), \theta_l\big)}{\partial \mathrm{LN}(x_l)}}_{A_l^{\mathrm{Pre-LN}}} \cdot \underbrace{\frac{\partial \mathrm{LN}(x_l)}{\partial x_l}}_{B_l^{\mathrm{Pre-LN}}}. \tag{4}$$

In terms of $B_l^{\mathrm{Pre-LN}}$, we consider two normalization methods: LayerNorm and RMSNorm, with their corresponding Jacobian matrices (w.r.t. the input vector $x$) denoted by $B^{\mathrm{LN}}(x)$ and $B^{\mathrm{RMS}}(x)$. For RMSNorm, we have Theorem 4.1.

**Theorem 4.1** (Spectral Norm of RMSNorm Jacobian). *Let* $\mathrm{RMSNorm}(x) := \frac{x}{\|x\|_{\mathrm{RMS}}+\epsilon}$, *where* $\|x\|_{\mathrm{RMS}} = \sqrt{\frac{1}{d}\sum_{i=1}^{d} x_i^2}$, *then the spectral norm of the* RMSNorm *Jacobian satisfies:*

$$\big\|B^{\mathrm{RMS}}(x)\big\|_2 = \frac{1}{\|x\|_{\mathrm{RMS}}+\epsilon}. \tag{5}$$

*Proof.* Given $\mathrm{RMSNorm}(x) := \frac{x}{\|x\|_{\mathrm{RMS}}+\epsilon}$, its Jacobian w.r.t. $x$ is

$$B^{\mathrm{RMS}}(x) = \frac{1}{\|x\|_{\mathrm{RMS}}+\epsilon}\Big[I - \frac{x\,x^T}{d\,(\|x\|_{\mathrm{RMS}}+\epsilon)^2}\Big], \tag{6}$$

where $d$ is the model dimension. Denote by $\alpha := \frac{1}{d\,(\|x\|_{\mathrm{RMS}}+\epsilon)^2}$. We have

$$B^{\mathrm{RMS}}(x) = \frac{1}{\|x\|_{\mathrm{RMS}}+\epsilon}\left[I - \alpha\,xx^T\right]. \tag{7}$$

Note that the matrix $\left[I - \alpha\,xx^T\right]$ has:

- $x$-orthogonal subspace ($\dim = d-1$): each vector $\mathbf{v}$ orthogonal to $x$ is an eigenvector with eigenvalue 1.

- the direction of $x$ has an eigenvalue $1 - \alpha\,\|x\|^2$, which is at most 1.

Hence its operator norm (largest singular value) is 1. Multiplying by the scalar $\frac{1}{\|x\|_{\mathrm{RMS}}+\epsilon}$ gives the final spectral norm:

$$\|B^{\mathrm{RMS}}(x)\|_2 = \frac{1}{\|x\|_{\mathrm{RMS}}+\epsilon}. \tag{8}$$

For LayerNorm, we have Theorem 4.2. □

**Theorem 4.2** (Spectral Norm of LayerNorm Jacobian). *Let* $\mathrm{LayerNorm}(x) := \frac{x-\mu\mathbf{1}}{\sqrt{\sigma^2+\epsilon}}$, *where* $\mu = \frac{1}{d}\sum_{i=1}^{d} x_i, \sigma^2 = \frac{1}{d}\sum_{i=1}^{d}(x_i - \mu)^2$, *then the spectral norm of the* LayerNorm *Jacobian satisfies:*

$$\|B^{\mathrm{LN}}(x)\|_2 = \frac{1}{\sqrt{\sigma^2+\epsilon}} \times \|M\|_2 = \frac{1}{\sqrt{\sigma^2+\epsilon}}. \tag{9}$$

*Proof.* Given $\mathrm{LayerNorm}(x) = \frac{x-\mu\mathbf{1}}{\sqrt{\sigma^2+\epsilon}}$, its Jacobian w.r.t. $x$ is

$$B^{\mathrm{LN}}(x) = \frac{1}{\sqrt{\sigma^2+\epsilon}}M, \tag{10}$$

where

$$M = I - \frac{1}{d}\mathbf{1}\mathbf{1}^T - \frac{(x-\mu\mathbf{1})(x-\mu\mathbf{1})^T}{d\,(\sigma^2+\epsilon)}. \tag{11}$$

Notice that $M$ is a rank-2 modification of the identity. On any vector orthogonal both to $\mathbf{1}$ and $(x-\mu\mathbf{1})$, the matrix $M$ acts as the identity. Meanwhile, in the 2D subspace spanned by $\{\mathbf{1}, x-\mu\mathbf{1}\}$, we have a certain negative correction. The largest singular value of $M$ is still 1 (or less), so we conclude

$$\|M\|_2 = 1. \tag{12}$$

Hence,

$$\|B^{\mathrm{LN}}(x)\|_2 = \frac{1}{\sqrt{\sigma^2+\epsilon}} \times \|M\|_2 = \frac{1}{\sqrt{\sigma^2+\epsilon}}. \tag{13}$$

□

Following the proof of Xiong et al. (2020) and Takase et al. (2023) with the assumption that $x$ follow a normal distribution. Therefore, we have $\|x\|_{\mathrm{RMS}}^2 = \sigma^2 + \mu^2$. According to Theorem 4.1 and Theorem 4.2, we can derive $\|B^{\mathrm{RMS}}(x)\|_2 \leq \|B^{\mathrm{LN}}(x)\|_2$. In practice, we usually observe $\sigma^2 > 1$ in later training stage Li et al. (2024a). Therefore, we have

$$\|B^{\mathrm{RMS}}(x)\|_2 \leq \|B^{\mathrm{LN}}(x)\|_2 < 1. \tag{14}$$

## 4.1 Larger Gradients in Shallower Layers

For the Pre-LN model, we have:

$$\begin{aligned}
\frac{\partial\mathcal{L}}{\partial\theta_l} &= \frac{\partial\mathcal{L}}{\partial x_{l+1}}\frac{\partial x_{l+1}}{\partial\theta_l} \\
&= \frac{\partial\mathcal{L}}{\partial x_L}\left(\prod_{k=l+1}^{L-1}\frac{\partial x_{k+1}}{\partial x_k}\right)\frac{\partial x_{l+1}}{\partial\theta_l} \\
&= \frac{\partial\mathcal{L}}{\partial x_L}\left(\prod_{k=l+1}^{L-1}\left(I + A_k^{\mathrm{Pre-LN}}B_k^{\mathrm{Pre-LN}}\right)\right)\frac{\partial x_{l+1}}{\partial\theta_l}.
\end{aligned} \tag{15}$$

According to Equation (15), the norm of the gradient factor can be upper bounded by

$$\|\frac{\partial \mathcal{L}}{\partial \theta_l}\| \le \|\frac{\partial \mathcal{L}}{\partial x_L}\| \prod_{k=l+1}^{L-1} \left( \|I + A_k^{\text{Pre-LN}} B_k^{\text{Pre-LN}}\| \right) \|\frac{\partial x_{l+1}}{\partial \theta_l}\|. \tag{16}$$

According to Equation (4), the derivative of the residual connection in Pre-LN is decoupled from the term associated with the derivative of LN, which prevents the vanishing gradient problem in early layers. According to Theorem 4.1 and Theorem 4.2, we have demonstrated that $\|B^{\text{RMS}}(x)\|_2 \le \|B^{\text{LN}}(x)\|_2 < 1$, i.e., $\|B_k^{\text{Pre-LN}}\| < 1$. Therefore, when $l$ is close to $L$, we are more likely to have a $\prod_{k=l+1}^{L-1} \left( \|I + A_k^{\text{Pre-LN}} B_k^{\text{Pre-LN}}\| \right)$ close to $I$ compared to shallower layers, indicating that deep layers are less important than shallow layers due to their gradients do not contribute much. The above derivation theoretically explains why it is reasonable for us to prune the last several layers.

## 4.2 FINE-TUNING THE LAST REMAINING LAYERS AFTER PRUNING OUTPERFORMS LORA

When pruning the top (or last few) layers and freezing the rest, a key decision is whether to apply LoRA (Hu et al., 2021) across all remaining layers or to fully fine-tune only the last layer. This scenario differs from standard fine-tuning settings, where LoRA applied to the full model is often preferred due to its parameter efficiency and ability to distribute adaptation across all layers.

From Equation (15), we observe that LoRA tends to have a more pronounced effect on shallower layers. However, in large pretrained models, lower-layer representations are already robust and broadly transferable to various downstream tasks, often requiring minimal modification. When the model is *not* pruned, fine-tuning the full model with LoRA is generally more effective since the entire network participates in adaptation. In contrast, after pruning, the distribution shift in the *last remaining layer* is more severe because it must align the modified feature space with the new output distribution. If LoRA exerts a weaker influence on deeper (remaining) layers, the final alignment with the task output may still be suboptimal. By fully fine-tuning *only* the last remaining layer, we directly adapt the part of the model that most critically shapes the final output distribution. As a result, this approach can yield better alignment and higher performance than a distributed LoRA update across all remaining layers, particularly in scenarios where deeper layers undergo significant pruning-induced shifts.

## 5 OBTAINING THE BEST PRUNED MODELS

In the previous sections, we have gained some valuable non-trivial practices and insights on LLM layer pruning through systematic experiments and provided theoretical proofs that underpin its feasibility. Herein, we use these practices to obtain the **Llama-3.1-6.3B-It-Alpaca** and **Llama-3-6.3B-Alpaca** models and compare their performance against pruned models obtained by various state-of-the-art LLM pruning methods, including ShortGPT (Men et al., 2024), Shortened LLaMA (Kim et al., 2024), SLEB (Song et al., 2024), PIP (Cao et al., 2025), LLM-Pruner (Ma et al., 2023), SliceGPT (Ashkboos et al., 2024), LaCo (Yang et al., 2024c), DeeperLayers Gromov et al. (2024), FINERCUT (Zhang et al., 2024b), LLM-Streamline (Chen et al., 2024) and GRASP (Liu et al., 2025).

Specifically, Llama-3.1-6.3B-It-Alpaca and Llama-3-6.3B-Alpaca are obtained by pruning 8 layers of Llama-3.1-8B-It and Llama-3-8B using the reverse-order metric. The pruned models are then fine-tuned with partial-layer fine-tuning (*lm_head* + *last three layers*) on the Alpaca-cleaned dataset. As shown in Table 4, experimental results show that our Llama-3.1-6.3B-It-Alpaca and Llama-3-6.3B-Alpaca outperform all baselines in terms of average accuracy. Specifically, as for Llama-3.1-8B-It, our method achieves an average accuracy of 58.07%, which is 5.62% higher than the best existing baseline GRASP and 17.27% higher than the worst-performing method ShortGPT. As for Llama-3-8B, our method achieves an average accuracy of 56.33%, which is 2.36% higher than the best existing baseline FINERCUT and 19.45% higher than the worst-performing method ShortGPT. It is worth noting that SLEB (Song et al., 2024) is processed based on the inference-only approach without any fine-tuning, so we fine-tune the pruned model using LoRA on Alpaca-cleaned for a fair comparison. We also present the results of pruning without fine-tuning in Table K, where reverse-order pruning still outperforms SLEB, highlighting its superior performance even without additional fine-tuning. Finally, we provide the generation results of our Llama-3.1-6.3B-It-Alpaca

| Model | PR | Method | Benchmarks | | | | | | | | Avg Acc↑ |
|---|---|---|---|---|---|---|---|---|---|---|---|
| | | | PIQA↑ | HellaSwag↑ | OpenbookQA↑ | ARC-e↑ | ARC-c↑ | MMLU↑ | CMMLU↑ | WinoGrande↑ | |
| Llama-3.1-8B-It | 0% | Original Model | 0.7987 | 0.5910 | 0.3380 | 0.8190 | 0.5171 | 0.6804 | 0.5543 | 0.7372 | 0.6295 |
| | 25% | ShortGPT (BI) | 0.7176 | 0.4196 | 0.2020 | 0.6107 | 0.2841 | 0.2417 | 0.2494 | 0.5391 | 0.4080 |
| | 25% | Shortened LLaMA (PPL) | **0.7628** | 0.4931 | 0.2640 | **0.7290** | 0.3805 | 0.3367 | 0.2724 | 0.5793 | 0.4772 |
| | 25% | Shortened LLaMA (Taylor) | 0.7138 | 0.4964 | 0.2740 | 0.6848 | 0.4181 | 0.2861 | 0.2504 | **0.7135** | 0.4796 |
| | 25% | SLEB (w/ fine-tuning) | 0.7573 | 0.4973 | 0.2680 | 0.6970 | 0.3865 | 0.4305 | 0.3338 | 0.6385 | 0.5011 |
| | 20% | LLM-Pruner* | 0.7200 | 0.5460 | - | - | - | 0.2530 | 0.2500 | - | 0.4423 |
| | 20% | SliceGPT* | 0.6830 | 0.4750 | - | - | - | 0.2880 | 0.2480 | - | 0.4235 |
| | 20% | LaCo* | 0.6980 | 0.5570 | - | - | - | 0.2650 | 0.2520 | - | 0.4408 |
| | 20% | LLM-Streamline* | 0.7150 | 0.6110 | - | - | - | 0.4550 | 0.2940 | - | 0.5188 |
| | 20% | GRASP* | 0.7330 | 0.6270 | - | - | - | 0.4310 | 0.3070 | - | 0.5245 |
| | 25% | Llama-3.1-6.3B-It-Alpaca | 0.7383 | 0.5323 | **0.3080** | 0.7260 | 0.4684 | 0.6567 | 0.5515 | 0.6646 | **0.5807** |
| Llama-3-8B | 0% | Original Model | 0.7965 | 0.6014 | 0.3480 | 0.8005 | 0.4983 | 0.6212 | 0.4752 | 0.7332 | 0.6093 |
| | 25% | ShortGPT (BI) | 0.6997 | 0.4095 | 0.1980 | 0.5825 | 0.2662 | 0.2310 | 0.2522 | 0.5194 | 0.3688 |
| | 25% | Shortened LLaMA (PPL) | **0.7612** | 0.4952 | 0.2880 | 0.7024 | 0.3669 | 0.2734 | 0.2546 | 0.5493 | 0.4614 |
| | 25% | Shortened LLaMA (Taylor) | 0.7138 | 0.4919 | 0.2740 | 0.6595 | 0.4018 | 0.3270 | 0.3179 | 0.6882 | 0.4843 |
| | 25% | SLEB (w/ fine-tuning) | 0.7514 | 0.5026 | 0.2780 | 0.7071 | 0.3720 | 0.3115 | 0.2683 | 0.5967 | 0.3947 |
| | 20% | LLM-Pruner† | 0.6950 | 0.4260 | 0.2700 | 0.5230 | 0.2950 | - | - | 0.6240 | 0.4722 |
| | 20% | SliceGPT† | 0.6540 | 0.3980 | 0.2380 | 0.5950 | 0.2940 | - | - | 0.6320 | 0.4685 |
| | 20% | PIP† | 0.6960 | 0.4470 | 0.2680 | 0.5790 | 0.3510 | - | - | **0.6940** | 0.5058 |
| | 25% | DeeperLayers‡ | 0.6260 | 0.3790 | 0.2820 | 0.3770 | 0.3120 | 0.3570 | - | 0.5340 | 0.4096 |
| | 25% | FINERCUT‡ | 0.7480 | **0.6010** | **0.3780** | 0.6220 | 0.3580 | 0.4120 | - | 0.6590 | 0.5397 |
| | 25% | Llama-3-6.3B-Alpaca | 0.7388 | 0.5476 | 0.3160 | **0.7218** | 0.4394 | 0.6179 | 0.4497 | 0.6748 | **0.5633** |

Table 4: Comparison with pruned models obtained through state-of-the-art LLM pruning methods. Results marked with ∗, † and ‡ are reported from Liu et al. (2025), Cao et al. (2025) and Zhang et al. (2024b), respectively. All other evaluations are run by us. PR denotes the pruning rate. The best results are marked in **boldface**, and the sub-optimal ones are underlined.

| Method | Benchmarks | | | | | | | | Avg Acc↑ |
|---|---|---|---|---|---|---|---|---|---|
| | PIQA↑ | HellaSwag↑ | OpenbookQA↑ | ARC-e↑ | ARC-c↑ | MMLU↑ | CMMLU↑ | WinoGrande↑ | |
| SLEB | **0.7916** | 0.5805 | 0.3360 | **0.8005** | 0.4974 | 0.5604 | 0.4125 | 0.7238 | 0.5878 |
| SliceGPT† | 0.6960 | 0.4440 | 0.3020 | 0.6970 | 0.4110 | - | - | 0.7210 | 0.5452 |
| ShortGPT† | 0.7600 | 0.5710 | 0.1900 | 0.7740 | 0.4790 | - | - | 0.6050 | 0.5632 |
| PIP† | 0.7860 | 0.5610 | 0.2960 | 0.7570 | 0.4530 | - | - | **0.7330** | 0.5977 |
| Our | 0.7612 | **0.5887** | **0.3580** | 0.7837 | **0.5606** | **0.7483** | **0.6288** | 0.6993 | **0.6411** |

Table 5: Comparison with pruned Llama-3-70B obtained through state-of-the-art LLM pruning methods under 20% pruning rate. Results marked with † are reported from Cao et al. (2025).

and the original Llama-3.1-8B-It in Table Q. It is evident that the sentences generated by Llama-3.1-6.3B-It-Alpaca are comparable to those produced by the original model. They exhibit fluency, relevance, and informativeness regarding the given topic.

**Generalization of the proposed method on large-scale models.** To further demonstrate the generalization of the proposed method, we conduct experiments on the LLaMA-3-70B model. Specifically, we compare our method with SliceGPT (Ashkboos et al., 2024), ShortGPT (Men et al., 2024), SLEB (Song et al., 2024) and PIP (Cao et al., 2025) under 20% pruning rate. As shown in Table 5, our method achieves the highest overall performance with an average accuracy of 64.11%, outperforming the strongest prior baseline (PIP) by 4.34%.

**Varying Pruning Ratios.** In this subsection, we explore the performance of our method at different pruning ratios. We conduct experiments on Llama-3.1-8B, controlling the pruning ratios at approximately 10%, 20% and 35% and compare our method with ShortGPT, LaCo, SliceGPT, 2SSP (Sandri et al., 2025) and COMPACT (Kwek & Yin, 2025). All benchmark experimental results are reported from Kwek & Yin (2025). Since Kwek & Yin (2025) primarily tests datasets such as MMLU, PIQA, WinoGrande, ARC-e, and ARC-c, our comparisons also focus on these evaluation tasks to ensure that the comparisons are fair. The experimental results are shown in Table 6. Our method achieves the highest average accuracy under each pruning rate (10%, 20%, 35%), further demonstrating the effectiveness of our method.

**Joint Pruning and Quantization.** In this subsection, we carry out joint pruning and quantization of the model, to see if their effects can be combined. We follow the setting of SparseGPT (Frantar & Alistarh, 2023) and MixGPT (Shao et al., 2024), where pruning is applied prior to quantization. Specifically, we apply 4-bit GPTQ (Frantar et al., 2022) quantization on the Llama-3.1-6.3B-It-

| PR | Method | MMLU↑ | PIQA↑ | WinoGrande↑ | ARC-e↑ | ARC-c↑ | Avg Acc↑ |
|---|---|---|---|---|---|---|---|
| 10.86% | ShortGPT | 0.5800 | 0.7750 | 0.7020 | 0.7120 | 0.4740 | 0.6486 |
| 10.86% | LaCo | 0.5880 | 0.7630 | 0.7230 | 0.7370 | 0.4890 | 0.6600 |
| 10.16% | SliceGPT | 0.4390 | 0.7100 | 0.6720 | 0.6620 | 0.3940 | 0.5754 |
| 10.86% | 2SSP | 0.5410 | **0.7970** | 0.7160 | 0.7160 | 0.4680 | 0.6476 |
| 10% | COMPACT | 0.5960 | 0.7840 | **0.7370** | 0.7490 | **0.5030** | 0.6738 |
| **12.50%** | Ours | **0.6785** | 0.7742 | 0.6993 | **0.7656** | 0.4949 | **0.6825** |
| 19.02% | ShortGPT | 0.5860 | 0.7160 | 0.6840 | 0.5830 | 0.4220 | 0.5982 |
| 19.02% | LaCo | 0.2410 | 0.7240 | 0.5530 | 0.5110 | 0.2920 | 0.4642 |
| 20.12% | SliceGPT | 0.2450 | 0.6190 | 0.6180 | 0.4900 | 0.3030 | 0.4550 |
| 19.99% | 2SSP | 0.3740 | **0.7680** | 0.6840 | 0.6180 | 0.3810 | 0.5650 |
| 20% | COMPACT | 0.5070 | 0.7590 | **0.7010** | 0.6600 | 0.4280 | 0.6110 |
| **21.88%** | Ours | **0.6378** | 0.7432 | 0.6654 | **0.7445** | **0.4744** | **0.6531** |
| **35.31%** | ShortGPT | 0.2320 | 0.5720 | 0.5910 | 0.3690 | 0.2970 | 0.4122 |
| **35.31%** | LaCo | 0.2310 | 0.5880 | 0.5310 | 0.3150 | 0.2730 | 0.3876 |
| 35.16% | SliceGPT | 0.2300 | 0.5500 | 0.5430 | 0.3720 | 0.2350 | 0.3860 |
| 34.77% | 2SSP | 0.2530 | 0.6870 | 0.5930 | 0.4430 | 0.2710 | 0.4494 |
| 34.99% | COMPACT | 0.3590 | 0.7060 | 0.6330 | 0.4840 | 0.3080 | 0.4980 |
| 34.38% | Ours | **0.6650** | **0.7067** | **0.6772** | **0.6705** | **0.4198** | **0.6278** |

Table 6: Comparison with pruned Llama-3.1-8B obtained through state-of-the-art LLM pruning methods under different pruning rates. The best results are marked in **boldface**.

| Method | Memory↓ | PIQA↑ | HellaSwag↑ | OpenbookQA↑ | ARC-e↑ | ARC-c↑ | MMLU↑ | CMMLU↑ | WinoGrande↑ | Avg Acc↑ |
|---|---|---|---|---|---|---|---|---|---|---|
| Llama-3.1-6.3B-It-Alpaca | 11988.40 MB | 0.7383 | 0.5323 | 0.3080 | 0.7260 | 0.4684 | 0.6567 | 0.5515 | 0.6646 | 0.5807 |
| Llama-3.1-6.3B-It-Alpaca+GPTQ | 1326.93 MB | 0.7193 | 0.5090 | 0.2860 | 0.6822 | 0.3985 | 0.6093 | 0.4571 | 0.6267 | 0.5360 |

Table 7: Zero-shot performance of quantized Llama-3.1-6.3B-It-Alpaca.

Alpaca. As shown in Table 7, after applying GPTQ for quantization, the model's memory footprint decreased significantly from $11988.40$ MB to $1326.93$ MB, a reduction of over $89\%$. Although this extreme compression leads to a performance degradation, the significant reduction in memory footprint and acceptable performance retention strongly demonstrate that our layer pruning and quantization, used in combination as orthogonal techniques, are highly effective.

**Performance comparison on GSM8K.** To further illustrate the reasoning capability of the pruned model obtained by our method, we evaluate the performance of pruned models obtained by various pruning methods on GSM8K (Cobbe et al., 2021) using Llama-3.1-8B. All benchmark results are reported from Kwek & Yin (2025). As shown in Table 8, despite the high pruning rate of our method, the accuracy is significantly higher than all baselines, reaching $29.19\%$. This further demonstrates the effectiveness of our proposed method.

| Method | Pruning Rate | Acc (%) |
|---|---|---|
| ShortGPT | 19.02% | 0.60 |
| LaCo | 19.02% | 0.40 |
| SliceGPT | 20.12% | 0.00 |
| 2SSP | 19.99% | 4.30 |
| COMPACT | 20.00% | 10.80 |
| Ours | 25% | **29.19** |

Table 8: Performance comparison on GSM8K.

# 6 CONCLUSION

In this paper, we revisit LLM layer pruning, focusing on pruning metrics and fine-tuning methods. From these efforts, we have developed a practical list of best practices for LLM layer pruning. These pruning strategies are further supported by theoretical analyses based on the gradient flow. Finally, we use these practices and insights to guide the pruning of Llama-3.1-8B-Instruct, Llama-3-8B and Llama-3-70B. Extensive experiments demonstrate that our pruning method outperforms various state-of-the-art pruning methods. Finally, we hope our work will inform best practices for pruning LLMs in real-world applications (Li et al., 2024b; Shen & Zhang, 2025; Wang et al., 2025; Liu et al., 2026; Feng et al., 2026; Lu et al., 2026; Zhao et al., 2026).

**Acknowledgments.** This work was supported by National Key R&D Program of China under Grant 2025YFA1510900, by the Baima Lake Laboratory Joint Fund of Zhejiang Provincial Natural Science Foundation of China under Grant LBMHZ25F020002, by the Key R&D Program of Zhejiang under Grant 2024C01025, by Key Technology Research and Development Program Project of Hangzhou under Grant 2024SZD1A23 and 2025SZD1A41, by the National Natural Science Foundation of China under Grant 62373325 and by the Zhejiang Provincial Natural Science Foundation of China under Grant LY21F030016.

**Ethics Statement.** In this paper, we carefully consider ethical concerns related to our research and ensure that all methodologies and experimental designs adhere to ethical standards. Our study focuses on layer pruning to enhance the efficiency of LLMs and reduce computational resource requirements, thereby promoting sustainable AI development. Furthermore, all models and datasets used in our research are sourced from publicly available and accessible origins, ensuring no infringement on intellectual property or personal privacy.

**Reproducibility Statement.** The authors have made great efforts to ensure the reproducibility of the empirical results reported in this paper. Firstly, the experiment settings, evaluation metrics, and datasets are described in detail in the main text. Secondly, the code to reproduce the results is available at `https://anonymous.4open.science/r/Navigation-LLM-layer-pruning-DEB7`.

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

APPENDIX

# A PROOFS

This proof provides a theoretical analysis, which starts from the chain rule with respect to model parameters, to explain why, in very deep **Pre-LN** (pre-layer normalization) Transformers, the performance gain from deeper layers might become marginal. The analysis can also be generalized to prove why, in very deep **Post-LN** (post-layer normalization) Transformers, the performance gain from shallow layers might become marginal. For simplicity, we consider a single residual path per layer, which contains a LayerNorm operator and a learnable sub-network $F$. We omit multi-head attention and FFN specifics to maintain clarity.

## A.1 MODEL STRUCTURE AND NOTATION

Consider an $L$-layer network (a simplified Transformer), where all layers follow *Pre-LN* residual blocks (or *Post-LN* residual blocks). Let:

- $x_0$ be the initial input (e.g., an embedding or the output of a preceding network).
- $x_l \in \mathbb{R}^d$ denote the input to layer $l$, with $l = 0, 1, \ldots, L - 1$.
- $x_{l+1}$ denote the output of layer $l$.

A Pre-LN residual block has the form:

$$x_{l+1} = x_l + F\Big(\text{LN}(x_l); \theta_l\Big),$$

where:

- $\text{LN}(\cdot)$ is LayerNorm,
- $F(\,\cdot\,; \theta_l)$ is the learnable sub-network at layer $l$ (e.g. including parameters for attention, feed-forward, etc.),
- $\theta_l$ collects all parameters at layer $l$,
- The residual part is represented by $x_l$.

Similarly, a Post-LN residual block has the form:

$$x_{l+1} = \text{LN}\Big(x_l + F(x_l; \theta_l)\Big).$$

The final layer's output $x_L$ goes through a Head function to produce a prediction $y$, which is used to compute the loss $\mathcal{L}(y, \text{target})$, where $\text{target}$ stands for the next token. Denote the final prediction by

$$y = \text{Head}\big(x_L, \theta_{\text{head}}\big).$$

Our goal is to examine the gradient

$$\frac{\partial \mathcal{L}}{\partial \theta_l} \quad \text{for each layer } l = 0, 1, \ldots, L - 1.$$

## A.2 CHAIN RULE: GRADIENT

According to the standard backpropagation (chain rule):

$$\frac{\partial \mathcal{L}}{\partial \theta_l} = \underbrace{\frac{\partial \mathcal{L}}{\partial x_{l+1}}}_{\text{incoming gradient}} \cdot \underbrace{\frac{\partial x_{l+1}}{\partial \theta_l}}_{\text{layer w.r.t. parameters}},$$

where

- $\dfrac{\partial x_{l+1}}{\partial \theta_l}$ describes how the layer output $x_{l+1}$ depends on the layer parameters $\theta_l$.

- $\dfrac{\partial \mathcal{L}}{\partial x_{l+1}}$ is the "incoming" gradient from deeper layers (from layer $l + 1$ to $L$ and finally to the loss). By chain rule:

$$\frac{\partial \mathcal{L}}{\partial x_{l+1}} = \frac{\partial \mathcal{L}}{\partial x_L} \cdot \frac{\partial x_L}{\partial x_{l+1}} = \frac{\partial \mathcal{L}}{\partial x_L} \cdot \prod_{k=l+1}^{L-1} \frac{\partial x_{k+1}}{\partial x_k}.$$

To understand the gradient flow through layer $l$, we need to analyze both above factors.

### A.3  SINGLE-LAYER JACOBIAN

Recall the Pre-LN block:

$$x_{l+1} = x_l + F\Big(\mathrm{LN}(x_l); \theta_l\Big).$$

Taking the derivative w.r.t. $x_l$, we get

$$\frac{\partial x_{l+1}}{\partial x_l} = I + \underbrace{\frac{\partial F\big(\mathrm{LN}(x_l), \theta_l\big)}{\partial \mathrm{LN}(x_l)}}_{A_l^{\mathrm{Pre-LN}}} \cdot \underbrace{\frac{\partial \mathrm{LN}(x_l)}{\partial x_l}}_{B_l^{\mathrm{Pre-LN}}}.$$

Denote

$$A_l^{\mathrm{Pre-LN}} = \frac{\partial F(\mathrm{LN}(x_l), \theta_l)}{\partial \mathrm{LN}(x_l)}, \quad B_l^{\mathrm{Pre-LN}} = \frac{\partial \mathrm{LN}(x_l)}{\partial x_l}.$$

Thus,

$$\boxed{J_l^{\mathrm{Pre-LN}} \equiv \frac{\partial x_{l+1}}{\partial x_l} = I + A_l^{\mathrm{Pre-LN}} B_l^{\mathrm{Pre-LN}}.}$$

Similarly, for the Post-LN block:

$$x_{l+1} = \mathrm{LN}\Big(x_l + F(x_l; \theta_l)\Big).$$

Taking the derivative w.r.t. $x_l$, we get

$$\frac{\partial x_{l+1}}{\partial x_l} = \underbrace{\frac{\partial \mathrm{LN}\big(x_l + F(x_l; \theta_l)\big)}{\partial(x_l + F(x_l; \theta_l))}}_{B_l^{\mathrm{Post-LN}}} \cdot \underbrace{\left(I + \frac{\partial F(x_l)}{\partial x_l}\right)}_{A_l^{\mathrm{Post-LN}}}.$$

Denote

$$B_l^{\mathrm{Post-LN}} = \frac{\partial \mathrm{LN}\big(x_l + F(x_l; \theta_l)\big)}{\partial(x_l + F(x_l; \theta_l))},$$

$$A_l^{\mathrm{Post-LN}} = \left(I + \frac{\partial F(x_l)}{\partial x_l}\right).$$

Thus,

$$\boxed{J_l^{\mathrm{Post-LN}} \equiv \frac{\partial x_{l+1}}{\partial x_l} = B_l^{\mathrm{Post-LN}} A_l^{\mathrm{Post-LN}}.}$$

In terms of $B_l$, we consider two normalization methods: *LayerNorm* and *RMSNorm*, with their corresponding Jacobian matrices (w.r.t. the input vector $x$) denoted by $B^{\mathrm{LN}}(x)$ and $B^{\mathrm{RMS}}(x)$.

The RMSNorm is defined as follows.

$$\mathrm{RMSNorm}(x) \equiv \frac{x}{\|x\|_{\mathrm{RMS}} + \epsilon},$$

where $\|x\|_{\mathrm{RMS}} = \sqrt{\frac{1}{d} \sum_{i=1}^{d} x_i^2}$. Thus its Jacobian w.r.t. $x$ is

$$B^{\mathrm{RMS}}(x) \equiv \frac{\partial \left(\mathrm{RMSNorm}(x)\right)}{\partial x}$$

$$= \frac{1}{\|x\|_{\mathrm{RMS}} + \epsilon}\left[I - \frac{x\,x^T}{d\,(\|x\|_{\mathrm{RMS}} + \epsilon)^2}\right],$$

where $d$ is the model dimension.

Consider $\|B^{\mathrm{RMS}}(x)\|_2$, the maximum singular value. Denote by $\alpha := \frac{1}{d\,(\|x\|_{\mathrm{RMS}}+\epsilon)^2}$. We have

$$B^{\mathrm{RMS}}(x) = \frac{1}{\|x\|_{\mathrm{RMS}} + \epsilon}\Big[I - \alpha\,xx^T\Big].$$

Note that the matrix $\big[I - \alpha\,xx^T\big]$ has:

- $x$-orthogonal subspace ($\dim = d-1$): each vector $\mathbf{v}$ orthogonal to $x$ is an eigenvector with eigenvalue 1.
- the direction of $x$ has an eigenvalue $1 - \alpha\,\|x\|^2$, which is at most 1.

Hence its operator norm (largest singular value) is 1. Multiplying by the scalar $\frac{1}{\|x\|_{\mathrm{RMS}}+\epsilon}$ gives the final spectral norm:

$$\|B^{\mathrm{RMS}}(x)\|_2 = \frac{1}{\|x\|_{\mathrm{RMS}} + \epsilon}.$$

The LayerNorm is defined as follows.

$$\mathrm{LayerNorm}(x) \equiv \frac{x - \mu\,\mathbf{1}}{\sqrt{\sigma^2 + \epsilon}},$$

where $\mu = \frac{1}{d}\sum_{i=1}^{d} x_i, \sigma^2 = \frac{1}{d}\sum_{i=1}^{d}(x_i - \mu)^2$. Its Jacobian w.r.t. $x$ is

$$B^{\mathrm{LN}}(x) \equiv \frac{\partial\,(\mathrm{LayerNorm}(x))}{\partial x} = \frac{1}{\sqrt{\sigma^2 + \epsilon}}M,$$

where

$$M = I - \frac{1}{d}\,\mathbf{1}\mathbf{1}^T - \frac{(x - \mu\,\mathbf{1})(x - \mu\,\mathbf{1})^T}{d\,(\sigma^2 + \epsilon)}.$$

Notice that $M$ is a rank-2 modification of the identity. On any vector orthogonal both to $\mathbf{1}$ and $(x - \mu\mathbf{1})$, the matrix $M$ acts as the identity. Meanwhile, in the 2D subspace spanned by $\{\mathbf{1}, x - \mu\mathbf{1}\}$, we have a certain negative correction. The largest singular value of $M$ is still 1 (or less), so we conclude

$$\|M\|_2 = 1.$$

Hence,

$$\|B^{\mathrm{LN}}(x)\|_2 = \frac{1}{\sqrt{\sigma^2 + \epsilon}} \times \|M\|_2 = \frac{1}{\sqrt{\sigma^2 + \epsilon}}.$$

Note that $\|x\|_{\mathrm{RMS}}^2 = \sigma^2 + \mu^2$. We know $\|B^{\mathrm{RMS}}(x)\|_2 \le \|B^{\mathrm{LN}}(x)\|_2$. In practice, we usually observe $\sigma^2 > 1$ in later training stage Li et al. (2024a). Therefore, we have $\|B^{\mathrm{RMS}}(x)\|_2 \le \|B^{\mathrm{LN}}(x)\|_2 < 1$.

### A.4  MULTI-LAYER CONCATENATION

Checking the partial derivative w.r.t. $x_{l+1}$, we have:

$$\frac{\partial \mathcal{L}}{\partial x_{l+1}} = \frac{\partial \mathcal{L}}{\partial x_L} \cdot \prod_{k=l+1}^{L-1} \frac{\partial x_{k+1}}{\partial x_k} = \frac{\partial \mathcal{L}}{\partial x_L} \cdot \prod_{k=l+1}^{L-1} J_k.$$

For Pre-LN, we have

$$\begin{aligned}
\frac{\partial \mathcal{L}}{\partial \theta_l} &= \frac{\partial \mathcal{L}}{\partial x_{l+1}} \frac{\partial x_{l+1}}{\partial \theta_l} \\
&= \frac{\partial \mathcal{L}}{\partial x_L} \cdot \left(\prod_{k=l+1}^{L-1} J_k^{\mathrm{Pre-LN}}\right) \cdot \frac{\partial x_{l+1}}{\partial \theta_l} \\
&= \frac{\partial \mathcal{L}}{\partial x_L} \cdot \left(\prod_{k=l+1}^{L-1} \left(I + A_k^{\mathrm{Pre-LN}} B_k^{\mathrm{Pre-LN}}\right)\right) \cdot \frac{\partial x_{l+1}}{\partial \theta_l}.
\end{aligned}$$

| Fine-tuning Method | Model | Metric | Iteration steps | Benchmarks | | | | | | | | Avg Acc |
|---|---|---|---|---|---|---|---|---|---|---|---|---|
| | | | | PIQA | HellaSwag | OpenbookQA | ARC-e | ARC-c | MMLU | CMMLU | WinoGrande | |
| LoRA | Llama-3.1-8B-It | Reverse-order | one-shot | 0.7002+0.0107 | 0.4010+0.0049 | 0.2940+0.0204 | 0.6170+0.0100 | 0.3985+0.0143 | 0.6342+0.0039 | 0.5449±0.0045 | 0.6243±0.0136 | 0.5268 |
| | | | 1:4:8 | 0.7176±0.0105 | 0.4538±0.0050 | 0.2920±0.0204 | 0.6705±0.0096 | 0.4121±0.0144 | 0.6374±0.0039 | 0.5439±0.0045 | 0.6369±0.0135 | 0.5455 |
| | | | 1:1:8 | 0.7160±0.0105 | 0.4470±0.0050 | 0.2860±0.0202 | 0.6637±0.0097 | 0.4061±0.0144 | 0.6440±0.0039 | 0.5425±0.0045 | 0.6448±0.0135 | **0.5438** |
| | | Taylor | one-shot | 0.7138±0.0105 | 0.4964±0.0050 | 0.2740±0.0200 | 0.6848±0.0095 | 0.4181±0.0144 | 0.2861±0.0038 | 0.2504±0.0040 | 0.7135±0.0127 | 0.4796 |
| | | | 1:4:8 | 0.7149±0.0105 | 0.4991±0.0050 | 0.2480±0.0193 | 0.7071±0.0093 | 0.3951±0.0143 | 0.4676±0.0041 | 0.3480±0.0044 | 0.6709±0.0132 | **0.5063** |
| | | | 1:1:8 | 0.6921±0.0108 | 0.4728±0.0050 | 0.2140±0.0184 | 0.6675±0.0097 | 0.3891±0.0142 | 0.4576±0.0041 | 0.3511±0.0044 | 0.6519±0.0134 | 0.4870 |
| Partial-layer | Llama-3.1-8B-It | Reverse-order | one-shot | 0.7383±0.0103 | 0.5323±0.0050 | 0.3080±0.0207 | 0.7260±0.0092 | 0.4684±0.0146 | 0.6567±0.0038 | 0.5515±0.0045 | 0.6646±0.0133 | 0.5807 |
| | | | 1:1:8 | 0.7432±0.0102 | 0.5357±0.0050 | 0.2980±0.0205 | 0.7496±0.0089 | 0.4590±0.0146 | 0.6539±0.0038 | 0.5558±0.0045 | 0.6922±0.0130 | **0.5859** |
| | | Taylor | one-shot | 0.7345±0.0103 | 0.5290±0.0050 | 0.3020±0.0206 | 0.7399±0.0090 | 0.4360±0.0145 | 0.6277±0.0039 | 0.4763±0.0046 | 0.7151±0.0127 | **0.5701** |
| | | | 1:1:8 | 0.6300±0.0113 | 0.3553±0.0048 | 0.1760±0.0170 | 0.5177±0.0103 | 0.2756±0.0131 | 0.2611±0.0037 | 0.2557±0.0041 | 0.5312±0.0140 | 0.3753 |

Table A: Zero-shot performance of pruned models (25% pruning rate) using different pruning strategies. The best results are marked in **boldface**. "1:1:8" refers to an iterative pruning process where 1 layer is pruned at a time, and a total of 8 layers are pruned by the end of the process.

Due to the existence of $I$, the gradients will not vanish when a larger number of layers are concatenated. Besides, for deeper layers, when $l$ is close to $L$, we are more likely to have a $\prod_{k=l+1}^{L-1} \left( \|I + A_k^{\mathrm{Pre-LN}} B_k^{\mathrm{Pre-LN}}\| \right)$ than is close to $I$ than shallow layers, indicating that deep layers are less important than shallow layers due to their gradients do not contribute much.

For Post-LN, we have

$$\frac{\partial \mathcal{L}}{\partial \theta_l} = \frac{\partial \mathcal{L}}{\partial x_{l+1}} \frac{\partial x_{l+1}}{\partial \theta_l}$$

$$= \frac{\partial \mathcal{L}}{\partial x_L} \cdot \left( \prod_{k=l+1}^{L-1} J_l^{\mathrm{Post-LN}} \right) \cdot \frac{\partial x_{l+1}}{\partial \theta_l}$$

$$= \frac{\partial \mathcal{L}}{\partial x_L} \cdot \left( \prod_{k=l+1}^{L-1} B_l^{\mathrm{Post-LN}} A_l^{\mathrm{Post-LN}} \right) \cdot \frac{\partial x_{l+1}}{\partial \theta_l}.$$

We know $\|B_l^{\mathrm{Post-LN}}\| < 1$. From the definition of $A_l^{\mathrm{Post-LN}}$, we know $\|A_l^{\mathrm{Post-LN}}\|$ is usually close to 1. Then, we usually have $\|B_l^{\mathrm{Post-LN}} A_l^{\mathrm{Post-LN}}\| < 1$. This indicates that shallow layers, which are followed by a larger number of subsequent layers, tend to receive smaller gradients and may even experience gradient vanishing. Therefore, with Post-LN, shallow layers are not important.

## B  WILL ITERATIVE PRUNING OUTPERFORM ONE-SHOT PRUNING?

In this subsection, we provide insights into the optimal pruning strategy for LLMs. Although Muralidharan et al. (2024) have explored pruning strategies and concluded that iterative pruning offers no benefit, their study focuses on utilizing knowledge distillation (Hinton, 2015) for performance recovery. In contrast, this paper concentrates on layer pruning with LoRA and partial-layer fine-tuning, thereby broadening the scope of pruning strategies evaluated. We briefly introduce the one-shot pruning and iterative pruning:

**One-shot Pruning.** One-shot pruning scores once and prunes the model to a target prune ratio.

**Iterative Pruning.** Iterative pruning alternately processes the score-prune-update cycle until achieving the target prune ratio.

Specifically, we select Llama-3.1-8B-Instruct and Gemma2-2B-Instruct as the base models. For one-shot pruning, we prune 8 layers from the Llama-3.1-8B-Instruct and 6 layers from the Gemma2-2B-Instruct in a single step, guided by the reverse-order and taylor metrics. For iterative pruning with LoRA, we begin by scoring all layers using these metrics. Subsequently, we set the pruning step to 1 and 4 for Llama-3.1-8B-Instruct, and 1 and 3 for Gemma2-2B-Instruct. After each pruning step, we fine-tune the model with LoRA and merge LoRA weights back into the fine-tuned model. This score-prune-fine-tune-merge cycle is repeated until a total of 8 layers are pruned for Llama-3.1-8B-Instruct and 6 layers for Gemma2-2B-Instruct. For iterative pruning with partial-layer fine-tuning, we fine-tune the model using partial-layer fine-tuning (*lm_head + last three layers*) after each pruning step, and then repeat the score-prune-fine-tune cycle. To avoid the fine-tuned layers being pruned completely, we set the pruning step size to 1. All hyperparameter settings are the same as in Section 3.3. Experiments with iterative pruning of more layers are provided in Table N.

**Results.** By comparing the results of iterative and one-shot pruning in Table A and Table N, we find that unlike traditional CNN pruning, which often yields significant performance improvements through iterative pruning (Tan & Motani, 2020; He & Xiao, 2023), the iterative approach for LLMs may not provide the same benefits and can even lead to performance degradation. We believe that is because too much training causes the model to suffer from **catastrophic forgetting** (Zhai et al., 2024; Liu et al., 2024a), which is evidenced by Figure B. It visualizes the representational similarity of different pruning strategies. From this, we observe that different pruning strategies yield significantly different representations, highlighting the impact of each strategy on the model's learned features. Besides, iterative pruning requires more computational overhead than one-shot pruning, which is not cost-effective with limited performance gains.

> **Insight:** Considering both performance gain and computational overhead, iterative pruning has no benefit.

## C DIFFERENCES FROM TRADITIONAL LAYER PRUNING

Unlike traditional Deep Neural Networks (Szegedy et al., 2014; Simonyan & Zisserman, 2015; He et al., 2015; Dosovitskiy et al., 2021; Liu et al., 2021) (DNNs), typically trained for a single, specific task, LLMs are designed to handle a wide range of tasks and are structured with billions of parameters. These differences in model scale and task complexity fundamentally alter the challenges associated with layer pruning. For example, in traditional DNN layer pruning (Chen & Zhao, 2018; Wang et al., 2019; Lu et al., 2022; Tang et al., 2023; Guenter & Sideris, 2024), assessing the importance of each layer is relatively straightforward, as it is tied to a single task. In contrast, the parameters of LLMs are optimized across diverse tasks, complicating the evaluation of layer importance. Furthermore, traditional DNN pruning commonly involves full parameter fine-tuning after pruning, while LLMs often employ Parameter-Efficient Fine-Tuning (PEFT) techniques (Hu et al., 2021; Meng et al., 2024; Zhao et al., 2024b; Dettmers et al., 2024) such as Low-Rank Approximation (LoRA) (Hu et al., 2021) to accommodate their massive parameter space. Consequently, traditional DNN pruning methods may not adequately address the unique challenges posed by LLMs, highlighting the need for specialized pruning strategies.

## D EXPERIMENTAL DETAILS

For the PPL metric, we follow (Ma et al., 2023; Muralidharan et al., 2024) and use WikiText2 for calculation. Following (Ma et al., 2023), we randomly select 10 samples from BookCorpus (Zhu et al., 2015) to compute Taylor and BI, truncating each sample to a sequence length of 128. Unless otherwise specified, we utilize the Alpaca-cleaned (Taori et al., 2023) with LoRA to recover the performance. Uniformly, we set the training epoch to 2 and batch size to 64. All experiments are conducted on 2 NVIDIA A100 GPUs with 40 GB of memory and 4 NVIDIA RTX A5000 GPUs with 24 GB of memory.

### D.1 DATASETS DESCRIPTION

**MMLU (Hendrycks et al., 2021)** is a massive multitask dataset consisting of multiple-choice questions from various branches of knowledge. The dataset spans subjects in the humanities, social sciences, hard sciences, and other areas and covers 57 tasks including elementary mathematics, US history, computer science, law, and more.

**CMMLU (Li et al., 2023a)** is a comprehensive Chinese assessment suite specifically designed to evaluate the advanced knowledge and reasoning abilities of LLMs within the Chinese language and cultural context. It covers a wide range of subjects, comprising 67 topics that span from elementary to advanced professional levels. CMMLU includes subjects that require computational expertise, such as physics and mathematics, as well as disciplines within humanities and social sciences.

**PIQA (Bisk et al., 2020)** is a dataset for commonsense reasoning, and is created to investigate the physical knowledge of existing models in NLP.

| Dataset | Benchmarks | | | | | | | | Avg Acc |
|---|---|---|---|---|---|---|---|---|---|
| | PIQA | HellaSwag | OpenbookQA | ARC-e | ARC-c | MMLU | CMMLU | WinoGrande | |
| Alpaca-cleaned | 0.7383±0.0103 | 0.5323±0.0050 | 0.3080±0.0207 | 0.7260±0.0092 | 0.4684±0.0146 | 0.6567±0.0038 | 0.5515±0.0045 | 0.6646±0.0133 | 0.5807 |
| MMLU | 0.6012±0.0114 | 0.2714±0.0044 | 0.1700±0.0168 | 0.3430±0.0097 | 0.2457±0.0126 | 0.5888±0.0040 | 0.5266±0.0045 | 0.5856±0.0138 | 0.4165 |

Table B: The effect of SFT datasets on LLM layer pruning.

**HellaSwag (Zellers et al., 2019)** is a challenge dataset for evaluating commonsense NLI that is specially hard for state-of-the-art models.

**OpenBookQA (Mihaylov et al., 2018)** is a new kind of question-answering dataset modeled after open book exams for assessing human understanding of a subject.

**ARC-easy Clark et al. (2018)** and **ARC-challenge Clark et al. (2018)** are two subsets of the AI2's Reasoning Challenge (ARC) dataset, which is a multiple-choice question-answering dataset containing questions from science exams from grade 3 to grade 9.

**WinoGrande (Sakaguchi et al., 2021)** is a new collection of 44k problems, inspired by Winograd Schema Challenge (Levesque et al., 2012), but adjusted to improve the scale and robustness against the dataset-specific bias. Formulated as a fill-in-a-blank task with binary options, the goal is to choose the right option for a given sentence which requires commonsense reasoning.

| Proportion | Benchmarks | | | | | | | | Avg Acc |
|---|---|---|---|---|---|---|---|---|---|
| | PIQA | HellaSwag | OpenbookQA | ARC-e | ARC-c | MMLU | CMMLU | WinoGrande | |
| 1.0 | 0.7383±0.0103 | 0.5323±0.0050 | 0.3080±0.0207 | 0.7260±0.0092 | 0.4684±0.0146 | 0.6567±0.0038 | 0.5515±0.0045 | 0.6646±0.0133 | 0.5807 |
| 0.8 | 0.7372±0.0103 | 0.5279±0.0050 | 0.3100±0.0207 | 0.7235±0.0092 | 0.4565±0.0146 | 0.6515±0.0038 | 0.5477±0.0045 | 0.6567±0.0133 | 0.5764 |
| 0.6 | 0.7399±0.0102 | 0.5242±0.0050 | 0.3140±0.0208 | 0.7100±0.0093 | 0.4497±0.0145 | 0.6551±0.0038 | 0.5487±0.0045 | 0.6582±0.0133 | 0.5747 |
| 0.4 | 0.7399±0.0102 | 0.5194±0.0050 | 0.3060±0.0206 | 0.7020±0.0094 | 0.4548±0.0146 | 0.6540±0.0038 | 0.5531±0.0045 | 0.6630±0.0133 | 0.574 |
| 0.2 | 0.7383±0.0103 | 0.5077±0.0050 | 0.2980±0.0205 | 0.6860±0.0095 | 0.4360±0.0145 | 0.6455±0.0038 | 0.5458±0.0045 | 0.6590±0.0133 | 0.5083 |

Table C: The number of samples used in SFT.

| Model | # Params | # MACs | Memory | Latency |
|---|---|---|---|---|
| Llama-3.1-6.3B-It-Alpaca | 6.29B | 368.65G | 23984MiB | 210.35s |

Table D: Statistics of Llama-3.1-6.3B-It-Alpaca.

# E  SENSITIVITY ANALYSIS

In this section, we conduct sensitivity analyses on the number of calibration samples, the choice of SFT dataset and various pruning rates for LLM layer pruning.

**The effect of number of calibration samples on LLM layer pruning.** It is worth noting that some data-driven layer pruning methods, such as BI and Taylor, rely upon calibration samples to generate layer activations. Therefore, we explore the effect of the number of calibration samples on pruning. Specifically, we calculate BI and Taylor metrics using 1, 5, 10, 30, and 50 calibration samples, prune 8 layers based on these metrics, finetune the pruned Llama-3.1-8B-Instruct models using LoRA, and evaluate their performance through lm-evaluation-harness package. For ease of comparison, we report the average accuracy on 8 datasets. For more details, see Table O. Besides, we report the model perplexity on the WikiText and Penn Treebank test set. As shown in Table J, we observe that the pruned models, obtained using varying numbers of calibration samples, do affect the model complexity and zero-shot performance, which suggests that *for data-driven pruning methods, performance stability should also be considered a key criterion when evaluating the quality of pruning technique.* It is worth noting that the reverse-order pruning method can complete the pruning without any calibration samples.

**The effect of SFT datasets on LLM layer pruning.** In the main text, we uniformly utilize Alpaca-cleaned (Taori et al., 2023) to fine-tune the pruned models. Herein, we aim to assess how fine-tuning a pruned model using different SFT datasets affects its performance. Specifically, we conduct experiments using the Reverse-order metric to remove 8 layers from the Llama-3.1-8B-Instruct and fine-tune the pruned model using *lm_head + last three layers* on MMLU (training set) (Hendrycks

et al., 2021). We set the maximum sequence length to 512 for MMLU. From Table B, we observe that *fine-tuning with different SFT datasets significantly affects the performance of pruned models*, highlighting the need for further exploration to identify the most suitable datasets for fine-tuning pruned models.

**The effect of different pruning rates on LLM layer pruning.** We investigate the impact of pruning the LLM at various pruning rates in Figure A. Specifically, we conduct one-shot pruning on Llama-3.1-8B-Instruct using reverse-order and taylor metrics and evaluate their effects on the model's performance with LoRA. All hyperparameter settings remain consistent with those in Section 3.3. As shown in Figure A, we observe that as the number of pruned layers increases, the performance of the model on all datasets tends to decrease and eventually converges. However, certain datasets, especially MMLU, CMMLU, and ARC-c, are highly sensitive to layer changes and degrade faster than others. Besides, after cutting off about 16 layers, the model was damaged, so we set the maximum pruning rate in the paper to 16 layers.

**The effect of different numbers of samples used in SFT.** According to Table B, the choice of SFT dataset significantly impacts model performance. To investigate further, we conduct additional experiments analyzing the effect of the number of samples used in SFT. Specifically, we use 20%, 40%, 60%, 80% and 100% of the Alpaca-cleaned dataset for partial fine-tuning. As shown in Table C, we find that the number of samples used in SFT indeed affects the performance of the pruned model. Using only 20% of the dataset leads to a substantial performance decline.

**Pruning without any subsequent training.** To demonstrate the effectiveness of our reverse-order pruning method, we compare it with SLEB, a well-known retraining-free pruning method in a setting without post-pruning training. To further illustrate the effectiveness of our method and provide a more comprehensive comparison, we also add LLM-Pruner to this "no-fine-tuning" comparative experiment. Specifically, we apply SLEB, LLM-Pruner, and Reverse-order algorithms to prune the Llama-3.1-8B-It and Llama-3-8B models, respectively, and then evaluate their zero-shot performance directly on benchmarks without any fine-tuning. As shown in Table K, even without any fine-tuning recovery, our simple reverse-order pruning significantly outperforms SLEB and LLM-Pruner in terms of average accuracy on Llama-3.1-8B-It and Llama-3-8B. This further demonstrates the effectiveness of our method, proving its strong competitiveness even as a pure, retraining-free compression technique.

**Generality of partial-layer fine-tuning.** In Table 2 of the main text, we have demonstrated the effectiveness of partial-layer fine-tuning on reverse-order pruning. To verify the robustness of our partial-layer fine-tuning method, we further apply it to a completely different pruning metric, Taylor, in Table H. Experimental results show that after using Taylor pruning, the average accuracy using traditional LoRA for fine-tuning is only $0.4796$. However, when we switch to our proposed partial-layer fine-tuning on the same model using Taylor pruning, the average accuracy significantly improves to $0.5701$. This experiment fully demonstrates that our partial-layer fine-tuning strategy is not only applicable to reverse-order pruning, but is also a universally effective performance recovery method.

**How many layers are appropriate for partial-layer fine-tuning?** In this subsection, we aim to extend partial-layer fine-tuning beyond the last three layers. Herein, we include experiments for more settings (e.g., last 4, 5, 6, 7, 8, 9, 10 layers) on Llama-3.1-8B-Instruct. As shown in Table E, as the number of fine-tuned layers increases, the average accuracy improves slightly, but the increase is very limited. However, during this process, GPU memory usage increases from $48.02$GB to $69.99$GB, and training time also increases from $7931.36$s to $10316.53$s. Therefore, considering both performance and computational cost, fine-tuning the last 3 layers can achieve most of the performance improvement with relatively low training resource consumption.

**Partial-layer Fine-tuning vs. Full Fine-tuning.** In this subsection, we explore the effects of the proposed partial-layer fine-tuning and full fine-tuning (training from scratch). Due to the extremely high cost of "training from scratch" on larger models, we choose TinyLlama-1.1B (Zhang et al., 2024a) as our test model. Specifically, we prune the last four layers. Then, we compare two approaches: fine-tuning only the last three layers of the remaining layers and training from scratch. As shown in Table F, fine-tuning the last 3 layers achieves an average score of $37.62$, while training from scratch achieves an average score of $37.84$. This experiment strongly demonstrates that our

| Type | MMLU | CMMLU | PIQA | OpenbookQA | WinoGrande | HellaSwag | ARC-e | ARC-c | Avg Acc | GPU memory | Training time (2 epoch) |
|---|---|---|---|---|---|---|---|---|---|---|---|
| *lm_head+last 3 layers* | 0.6567 | 0.5515 | 0.7383 | 0.3080 | 0.6646 | 0.5323 | 0.7260 | 0.4684 | 0.5807 | 48.02G | 7931.36s |
| *lm_head+last 4 layers* | 0.6487 | 0.5530 | 0.7405 | 0.2940 | 0.6685 | 0.5370 | 0.7285 | 0.4625 | 0.5791 | 49.50G | 8242.46s |
| *lm_head+last 5 layers* | 0.6429 | 0.5545 | 0.7448 | 0.3080 | 0.6811 | 0.5391 | 0.7420 | 0.4659 | 0.5848 | 51.49G | 8572.35s |
| *lm_head+last 6 layers* | 0.6414 | 0.5495 | 0.7476 | 0.3040 | 0.6898 | 0.5418 | 0.7462 | 0.4633 | 0.5855 | 55.18G | 8902.82s |
| *lm_head+last 7 layers* | 0.6487 | 0.5530 | 0.7405 | 0.2940 | 0.6685 | 0.5370 | 0.7285 | 0.4625 | 0.5791 | 58.89G | 9289.69s |
| *lm_head+last 8 layers* | 0.6445 | 0.5441 | 0.7497 | 0.2980 | 0.6859 | 0.5457 | 0.7572 | 0.4582 | 0.5854 | 62.59G | 9607.29s |
| *lm_head+last 9 layers* | 0.6487 | 0.5432 | 0.7519 | 0.2960 | 0.6835 | 0.5475 | 0.7529 | 0.4642 | 0.5860 | 66.30G | 9975.59s |
| *lm_head+last 10 layers* | 0.6576 | 0.5437 | 0.7470 | 0.3000 | 0.6748 | 0.5497 | 0.7521 | 0.4753 | 0.5875 | 69.99G | 10316.53s |

Table E: Zero-shot performance of pruned Llama-3.1-8B-It models using various fine-tuning methods under $25\%$ pruning rate (using reverse-order).

| Method | MMLU↑ | CMMLU↑ | PIQA↑ | OpenbookQA↑ | WinoGrande↑ | HellaSwag↑ | ARC-e↑ | ARC-c↑ | Avg Acc↑ |
|---|---|---|---|---|---|---|---|---|---|
| *lm head+last three layers* | 0.2497 | 0.2479 | 0.6480 | 0.1940 | 0.5541 | 0.3937 | 0.4520 | 0.2705 | 0.3762 |
| Full fine-tuning | 0.2478 | 0.2497 | 0.6670 | 0.1880 | 0.5667 | 0.3879 | 0.4714 | 0.2491 | 0.3784 |

Table F: Zero-shot performance of the pruned model using partial-layer and full fine-tuning.

method can achieve performance almost equivalent to training a model of the same size from scratch with a much lower computational cost than training from scratch.

**Perplexity of the pruned models.** Perplexity is an important metric for evaluating the language modeling capabilities of pruned models. Therefore, we evaluate the performance of various pruned models ($20\%$ pruning rate, Llama-3-8B) on WikiText-2. As shown in Section E, our method achieves the lowest perplexity value of $19.16$, significantly outperforming all other pruning methods. This further demonstrates that our proposed method can maximally preserve the general generative capabilities of the pruned model.

| Method | Wanda | LLM-Pruner | ShortGPT | OWL | GISP | Ours |
|---|---|---|---|---|---|---|
| Perplexity↓ | 29.92 | 23.21 | 118.62 | 29.49 | 24.18 | **19.16** |

Table G: Comparison of perplexity of models obtained by different pruning methods.

**Pruning Mixture-of-Experts (MoE) models.** To further demonstrate that our pruning algorithm can be applied to different model architectures, especially MoE models, we pruned the Mixtral 8x7B (Jiang et al., 2024) model. Specifically, we prune the last $5$ blocks of the Mixtral 8x7B and compare it with the MoE pruning baseline UMD (He et al., 2024) with the same pruning rate. As shown in Table H, our method achieves an average accuracy of $0.6673$, which is higher than UMD's $0.6558$. This further demonstrates that our method has good architectural generality: even in complex MoE models, our simple reverse pruning strategy can still outperform existing MoE pruning baselines and achieve better performance.

**Performance comparison on HumanEval.** To further illustrate the reasoning capability of the pruned model obtained by our method, we evaluate the performance of pruned models obtained by various pruning methods on HumanEval (Chen et al., 2021) using CodeQwen1.5-7B-Chat (Bai et al., 2023). We set the pruning rate to $20\%$ and compare it with ShortGPT, UIDL (Gromov et al., 2024), Linearity (Razzhigaev et al., 2024), SLEB, and LLM-Pruner. As shown in Table I, our method achieves the highest Pass1 at $48.72$, significantly exceeding ShortGPT ($42.68$) and LLM-Pruner ($15.85$). Finally, the superior performance in code benchmarks further validates the effectiveness and generality of our method.

**Statistics of Llama-3.1-6.3B-It-Alpaca.** Table D presents the statistic of Llama-3.1-6.3B-It-Alpaca, including parameters, MACs, memory requirements and latency. Following Ma et al. (2023), the statistical evaluation is conducted in inference mode, where the model is fed a sentence consisting of 64 tokens. The latency is tested under the test set of WikiText2 on a single NVIDIA A100 GPU.

**Limitations and Future Work.** In this paper, we focus primarily on layer pruning due to the straightforward nature of pruning layers in LLMs, where the input and output dimensions are identical. However, we plan to further investigate weight pruning (Sun et al., 2023; Frantar & Alistarh, 2023) and width pruning (Ma et al., 2023) in future experiments. Besides, in this paper, we gain insights by focusing solely on the upper bound of the $\ell_2$-norm of the gradient flow. We acknowl-

| Method | MMLU | PIQA | OpenbookQA | WinoGrande | HellaSwag | ARC-c | Avg Acc |
|--------|------|------|------------|------------|-----------|-------|---------|
| UMD | 0.6790 | 0.7930 | 0.4200 | 0.7430 | 0.7870 | 0.5130 | 0.6558 |
| Ours | 0.6957 | 0.7890 | 0.4321 | 0.7674 | 0.7981 | 0.5216 | 0.6673 |

Table H: Performance comparison of pruned MoE models.

| Method | ShortGPT | UIDL | Linearity | SLEB | LLM-Pruner | Ours |
|--------|----------|------|-----------|------|------------|------|
| HumanEval(Pass@1) | 42.68 | 0.00 | 0.00 | 20.73 | 15.85 | 48.72 |

Table I: Performance comparison on HumanEval.

edge that matrix multiplication is significantly more complex than a simple norm, particularly in the context of LLMs. A comprehensive analysis is beyond the scope of this paper and is left for future work. Besides, since quantization may alter gradient flow or activation distribution, making otherwise unimportant layers crucial. Therefore, we will investigate the impact of quantization on layer importance in the future.

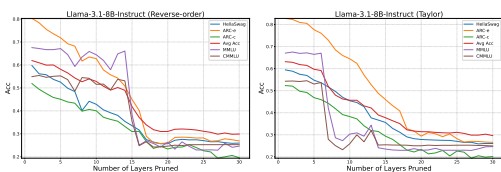

Figure A: The effect of different pruning rates on LLM layer pruning.

| | Verification | PPL on WikiText2 | | PPL on PTB | | Avg Acc | |
|---|---|---|---|---|---|---|---|
| | Metric | BI | Taylor | BI | Taylor | BI | Taylor |
| Calibration Samples | 1 | 51.06 | 65.43 | 90.97 | 94.35 | 0.40 | 0.36 |
| | 5 | 43.54 | 65.43 | 79.34 | 94.35 | 0.43 | 0.36 |
| | 10 | 53.53 | 65.43 | 101.64 | 94.35 | 0.41 | 0.36 |
| | 30 | 50.03 | 55.42 | 88.02 | 77.63 | 0.42 | 0.55 |
| | 50 | 59.73 | 55.42 | 103.19 | 77.63 | 0.41 | 0.55 |

Table J: The effect of number of calibration samples on LLM layer pruning. For more details, please refer to Table O.

| Model | Method | Benchmarks | | | | | | | | Avg Acc |
|---|---|---|---|---|---|---|---|---|---|---|
| | | PIQA | HellaSwag | OpenbookQA | ARC-e | ARC-c | MMLU | CMMLU | WinoGrande | |
| Llama-3.1-8B-It | SLEB (w/o fine-tuning) | 0.7252±0.0104 | 0.4415±0.0050 | 0.2380±0.0191 | 0.6423±0.0098 | 0.3166±0.0136 | 0.3396±0.0040 | 0.2756±0.0042 | 0.5888±0.0138 | 0.4192 |
| | LLM-Pruner (w/o fine-tuning) | 0.7356±0.0105 | 0.4518±0.0050 | 0.2760±0.0198 | 0.6713±0.0100 | 0.3473±0.0144 | 0.3413±0.0038 | 0.2704±0.0047 | 0.6559±0.0135 | 0.4687 |
| | Reverse-order (w/o fine-tuning) | 0.7002±0.0107 | 0.4021±0.0049 | 0.2920±0.0204 | 0.6178±0.0100 | 0.3993±0.0143 | 0.6346±0.0039 | 0.5458±0.0045 | 0.6251±0.0136 | **0.5271** |
| Llama-3-8B | SLEB (w/o fine-tuning) | 0.7111±0.0106 | 0.4401±0.0050 | 0.2280±0.0188 | 0.6014±0.0100 | 0.2807±0.0131 | 0.2674±0.0037 | 0.2502±0.0040 | 0.5683±0.0139 | 0.3689 |
| | LLM-Pruner (w/o fine-tuning) | 0.7220±0.0105 | 0.4578±0.0050 | 0.2840±0.0189 | 0.6338±0.0100 | 0.3319±0.0144 | 0.2700±0.0036 | 0.2612±0.0044 | 0.6125±0.0138 | 0.4467 |
| | Reverse-order (w/o fine-tuning) | 0.6921±0.0108 | 0.4035±0.0049 | 0.3040±0.0206 | 0.6014±0.0100 | 0.3720±0.0141 | 0.5603±0.0040 | 0.4216±0.0045 | 0.5975±0.0138 | **0.4940** |

Table K: Comparisons with retraining-free (pruning without fine-tuning) pruning method.

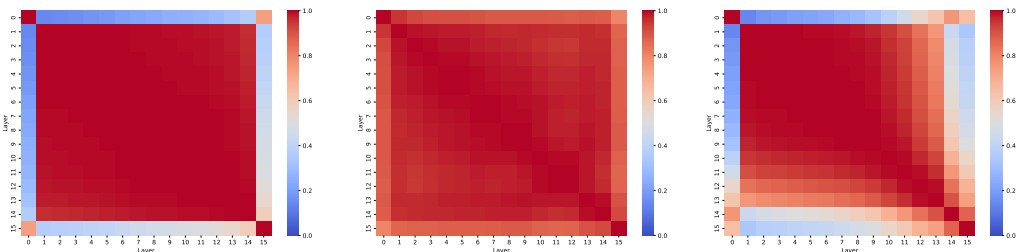

Figure B: Visualization of the layer similarity matrix of 16-layer pruned Llama-3.1-8B-It models (using Taylor) obtained by different pruning strategies. Left: one-shot pruning; Middle: iterative pruning with pruning step = 1; Right: iterative pruning with pruning step = 8.

| Model | Metric | Benchmarks | | | | | | | | Avg Acc |
|---|---|---|---|---|---|---|---|---|---|---|
| | | PIQA | HellaSwag | OpenbookQA | ARC-e | ARC-c | MMLU | CMMLU | WinoGrande | |
| Vicuna-7B-v1.5 | Dense | 0.7720±0.0098 | 0.5642±0.0049 | 0.3300±0.0210 | 0.7555±0.0088 | 0.4326±0.0145 | 0.4858±0.0040 | 0.3518±0.0044 | 0.6953±0.0129 | 0.5484 |
| | Random | 0.5773±0.0115 | 0.3083±0.0046 | 0.1560±0.0162 | 0.3775±0.0099 | 0.2176±0.0121 | 0.2650±0.0037 | 0.2542±0.0041 | 0.5067±0.0141 | 0.3328 |
| | PPL | **0.6572±0.0111** | **0.3524±0.0048** | **0.1940±0.0177** | **0.4971±0.0103** | 0.2406±0.0125 | 0.2361±0.0036 | 0.2510±0.0040 | 0.5328±0.0140 | **0.3702** |
| | Magnitude-l1 | 0.5239±0.0117 | 0.2585±0.0044 | 0.1400±0.0155 | 0.2635±0.0090 | 0.2184±0.0121 | 0.2295±0.0035 | 0.2527±0.0040 | 0.4893±0.0140 | 0.2970 |
| | Magnitude-l2 | 0.5245±0.0117 | 0.2590±0.0044 | 0.1300±0.0151 | 0.2656±0.0091 | 0.2210±0.0121 | 0.2293±0.0035 | 0.2512±0.0040 | 0.4791±0.0140 | 0.2950 |
| | BI | 0.5250±0.0117 | 0.2598±0.0044 | 0.1440±0.0157 | 0.2740±0.0092 | 0.1928±0.0115 | 0.2296±0.0035 | 0.2476±0.0040 | 0.4988±0.0141 | 0.2965 |
| | Taylor | 0.5283±0.0116 | 0.2585±0.0044 | 0.1300±0.0151 | 0.2572±0.0090 | 0.2167±0.0120 | 0.2614±0.0037 | 0.2513±0.0040 | 0.4901±0.0140 | 0.2992 |
| | Reverse-order | 0.5642±0.0116 | 0.2919±0.0045 | 0.1700±0.0168 | 0.3258±0.0096 | **0.2645±0.0129** | **0.4372±0.0041** | **0.3069±0.0043** | **0.5872±0.0138** | 0.3685 |
| Qwen1.5-7B | Dense | 0.7845±0.0096 | 0.5785±0.0049 | 0.3160±0.0208 | 0.7125±0.0093 | 0.4053±0.0143 | 0.5967±0.0039 | 0.7277±0.0039 | 0.6575±0.0133 | 0.5973 |
| | Random | 0.6409±0.0112 | **0.3268±0.0047** | **0.1940±0.0177** | **0.4617±0.0102** | 0.2261±0.0122 | 0.2321±0.0036 | 0.2529±0.0040 | 0.5083±0.0141 | **0.3553** |
| | PPL | **0.6529±0.0111** | 0.3233±0.0047 | 0.1700±0.0168 | 0.4360±0.0102 | 0.2099±0.0119 | 0.2297±0.0035 | 0.2541±0.0040 | 0.5225±0.0140 | 0.3498 |
| | Magnitude-l1 | 0.5452±0.0116 | 0.2690±0.0044 | 0.1280±0.0150 | 0.2837±0.0092 | 0.1962±0.0116 | 0.2548±0.0037 | 0.2479±0.0040 | 0.4862±0.0140 | 0.3013 |
| | Magnitude-l2 | 0.5348±0.0116 | 0.2651±0.0044 | 0.1520±0.0161 | 0.2858±0.0093 | 0.1843±0.0113 | **0.2659±0.0037** | 0.2519±0.0040 | 0.5059±0.0141 | 0.3057 |
| | BI | 0.6001±0.0114 | 0.2905±0.0045 | 0.1880±0.0175 | 0.4099±0.0101 | 0.2090±0.0119 | 0.2420±0.0036 | 0.2472±0.0040 | 0.4901±0.0140 | 0.3346 |
| | Taylor | 0.5223±0.0117 | 0.2540±0.0043 | 0.1460±0.0158 | 0.2403±0.0088 | 0.2176±0.0121 | 0.2393±0.0036 | 0.2478±0.0040 | 0.4854±0.0140 | 0.2941 |
| | Reverse-order | 0.5783±0.0115 | 0.3100±0.0046 | 0.1640±0.0166 | 0.3047±0.0094 | **0.2363±0.0124** | 0.2507±0.0037 | **0.2564±0.0041** | **0.5391±0.0140** | 0.3299 |
| Llama-3.1-8B-It | Dense | 0.8003±0.0093 | 0.5910±0.0049 | 0.3380±0.0212 | 0.8182±0.0079 | 0.5179±0.0146 | 0.6790±0.0038 | 0.5552±0.0045 | 0.7395±0.0123 | 0.6299 |
| | Random | 0.5588±0.0116 | 0.2730±0.0044 | 0.1280±0.0150 | 0.2826±0.0093 | 0.1903±0.0115 | 0.2406±0.0036 | **0.2555±0.0041** | 0.5020±0.0141 | 0.3039 |
| | PPL | **0.6643±0.0110** | **0.3548±0.0048** | **0.1960±0.0178** | **0.4718±0.0102** | 0.2483±0.0126 | 0.2394±0.0036 | 0.2446±0.0040 | 0.5454±0.0140 | 0.3706 |
| | Magnitude-l1 | 0.5316±0.0116 | 0.2576±0.0044 | 0.1360±0.0153 | 0.2572±0.0090 | 0.1980±0.0116 | 0.2344±0.0036 | 0.2526±0.0040 | 0.4933±0.0141 | 0.2951 |
| | Magnitude-l2 | 0.5316±0.0116 | 0.2576±0.0044 | 0.1360±0.0153 | 0.2572±0.0090 | 0.1980±0.0116 | 0.2344±0.0036 | 0.2526±0.0040 | 0.4933±0.0141 | 0.2951 |
| | BI | 0.5773±0.0115 | 0.2878±0.0045 | 0.1520±0.0161 | 0.3674±0.0099 | 0.1706±0.0110 | 0.2342±0.0036 | 0.2466±0.0040 | 0.5036±0.0141 | 0.3174 |
| | Taylor | 0.6088±0.0114 | 0.3288±0.0047 | 0.1660±0.0167 | 0.4318±0.0102 | 0.2790±0.0131 | 0.2310±0.0036 | 0.2534±0.0041 | 0.6093±0.0137 | 0.3635 |
| | Reverse-order | 0.6376±0.0112 | 0.3163±0.0046 | **0.1960±0.0178** | 0.4019±0.0101 | **0.3106±0.0135** | 0.2502±0.0036 | 0.2482±0.0040 | **0.6101±0.0137** | 0.3714 |

Table L: Zero-shot performance of the pruned models (50% pruning rate, fine-tuning using LoRA). "Avg Acc" denotes the average accuracy calculated among eight datasets. The best results are marked in **boldface**, and the sub-optimal ones are underlined.

| Model | Method | Layer | Benchmarks | | | | | | | | Avg Acc |
|---|---|---|---|---|---|---|---|---|---|---|---|
| | | | PIQA | HellaSwag | OpenbookQA | ARC-e | ARC-c | MMLU | CMMLU | WinoGrande | |
| Llama-3.1-8B-It | LoRA | - | 0.7138±0.0105 | 0.4964±0.0050 | 0.2740±0.0200 | 0.6848±0.0095 | 0.4181±0.0144 | 0.2861±0.0038 | 0.2504±0.0040 | 0.7135±0.0127 | 0.4796 |
| | QLoRA | - | 0.6496±0.0111 | 0.3260±0.0047 | 0.1820±0.0173 | 0.4520±0.0102 | 0.2969±0.0134 | 0.3425±0.0040 | 0.2627±0.0041 | 0.5793±0.0139 | 0.3864 |
| | Partial-layer | *lm_head only* | 0.6752±0.0109 | 0.3685±0.0048 | 0.2100±0.0182 | 0.5349±0.0102 | 0.3276±0.0137 | 0.4315±0.0041 | 0.3373±0.0044 | 0.6795±0.0109 | 0.4456 |
| | | *lm_head+last layer* | 0.7029±0.0107 | 0.4676±0.0050 | 0.2140±0.0184 | 0.6393±0.0099 | 0.3763±0.0142 | 0.5682±0.0041 | 0.4483±0.0046 | 0.6748±0.0132 | 0.5114 |
| | | *lm_head+last two layers* | 0.7252±0.0104 | 0.5173±0.0050 | 0.2800±0.0201 | 0.7104±0.0093 | 0.4232±0.0144 | 0.6058±0.0040 | 0.4659±0.0046 | 0.7040±0.0128 | 0.5540 |
| | | *lm_head+last three layers* | **0.7345±0.0103** | **0.5290±0.0050** | **0.3020±0.0206** | **0.7399±0.0090** | **0.4360±0.0145** | **0.6277±0.0039** | **0.4763±0.0046** | **0.7151±0.0127** | **0.5701** |

Table M: Zero-shot performance of the pruned models using various fine-tuning methods under 25% pruning rate (using Taylor metric). "Avg Acc" denotes the average accuracy calculated among eight datasets. The best results are marked in **boldface**, and the sub-optimal ones are underlined.

| Fine-tuning Method | Model | Method | Iteration steps | Benchmarks | | | | | | | | Avg Acc |
|---|---|---|---|---|---|---|---|---|---|---|---|---|
| | | | | PIQA | HellaSwag | OpenbookQA | ARC-e | ARC-c | MMLU | CMMLU | WinoGrande | |
| LoRA | Llama-3.1-8B-It | Reverse-order | one-shot | 0.6376±0.0112 | 0.3163±0.0046 | 0.1960±0.0178 | 0.4019±0.0101 | 0.3106±0.0135 | 0.2502±0.0036 | 0.2482±0.0040 | 0.6101±0.0137 | 0.3714 |
| | | | 1:8:16 | 0.6376±0.0112 | 0.3160±0.0046 | 0.1980±0.0178 | 0.3990±0.0100 | 0.3106±0.0135 | 0.2526±0.0037 | 0.2504±0.0040 | 0.6046±0.0137 | 0.3711 |
| | | | 1:1:16 | 0.6333±0.0112 | 0.3259±0.0047 | 0.2020±0.0180 | 0.4146±0.0101 | 0.2961±0.0133 | 0.2426±0.0036 | 0.2690±0.0041 | 0.5912±0.0138 | **0.3718** |
| | | Taylor | one-shot | 0.6088±0.0114 | 0.3288±0.0047 | 0.1660±0.0167 | 0.4318±0.0102 | 0.2790±0.0131 | 0.2310±0.0036 | 0.2534±0.0041 | 0.6093±0.0137 | **0.3635** |
| | | | 1:8:16 | 0.6230±0.0113 | 0.3516±0.0048 | 0.1480±0.0159 | 0.4604±0.0102 | 0.2355±0.0124 | 0.2541±0.0037 | 0.2546±0.0041 | 0.5312±0.0140 | 0.3573 |
| | | | 1:1:16 | 0.5430±0.0116 | 0.2692±0.0044 | 0.1580±0.0163 | 0.2921±0.0093 | 0.1937±0.0115 | 0.2334±0.0036 | 0.2481±0.0040 | 0.5091±0.0141 | 0.3058 |
| Partial-layer | Llama-3.1-8B-It | Reverse-order | one-shot | 0.6578±0.0111 | 0.4137±0.0049 | 0.2200±0.0185 | 0.5707±0.0102 | 0.3294±0.0137 | 0.3854±0.0040 | 0.3190±0.0043 | 0.6504±0.0134 | 0.4433 |
| | | | 1:1:16 | 0.6774±0.0109 | 0.4164±0.0049 | 0.2200±0.0185 | 0.5863±0.0101 | 0.3362±0.0138 | 0.4170±0.0041 | 0.3460±0.0044 | 0.6385±0.0135 | **0.4547** |
| | | Taylor | one-shot | 0.6649±0.0110 | 0.3985±0.0049 | 0.2100±0.0182 | 0.5581±0.0102 | 0.3251±0.0137 | 0.3054±0.0039 | 0.2876±0.0042 | 0.6212±0.0136 | **0.4214** |
| | | | 1:1:16 | 0.5876±0.0115 | 0.2813±0.0045 | 0.1300±0.0151 | 0.3986±0.0100 | 0.1980±0.0116 | 0.2508±0.0037 | 0.2502±0.0040 | 0.4957±0.0141 | 0.3240 |

Table N: Zero-shot performance of pruned models (50% pruning rate) using different pruning strategies. "Avg Acc" denotes the average accuracy calculated among eight datasets. The best results are marked in **boldface**. "1:1:12" refers to an iterative pruning process where 1 layer is pruned at a time, and a total of 12 layers are pruned by the end of the process.

| Model | Metric | Calibration Samples | Removed Layers | Benchmarks | | | | | | | | Avg Acc |
|---|---|---|---|---|---|---|---|---|---|---|---|---|
| | | | | PIQA | HellaSwag | OpenbookQA | ARC-e | ARC-c | MMLU | CMMLU | WinoGrande | |
| Llama-3.1-8B-Instruct | BI | 1 | 2,3,5,6,7,8,11,12 | 0.7029±0.0107 | 0.4167±0.0049 | 0.2060±0.0181 | 0.6136±0.0100 | 0.2739±0.0130 | 0.2362±0.0036 | 0.2512±0.0040 | 0.5225±0.0140 | 0.40 |
| | | 5 | 3,4,5,8,9,10,13,19 | 0.7236±0.0104 | 0.4400±0.0050 | 0.2420±0.0192 | 0.6730±0.0096 | 0.3311±0.0138 | 0.2524±0.0037 | 0.2553±0.0041 | 0.5485±0.0140 | 0.43 |
| | | 10 | 2,3,4,5,6,7,8,9 | 0.7176±0.0105 | 0.4196±0.0049 | 0.2020±0.0180 | 0.6107±0.0100 | 0.2841±0.0132 | 0.2417±0.0036 | 0.2494±0.0040 | 0.5391±0.0140 | 0.41 |
| | | 30 | 2,3,4,10,11,12,13,14 | 0.7209±0.0105 | 0.4328±0.0049 | 0.2040±0.0180 | 0.6414±0.0098 | 0.3259±0.0137 | 0.2500±0.0036 | 0.2576±0.0041 | 0.5517±0.0140 | 0.42 |
| | | 50 | 2,3,4,5,6,7,10,13 | 0.7100±0.0106 | 0.4091±0.0049 | 0.2180±0.0185 | 0.6221±0.0099 | 0.2875±0.0132 | 0.2492±0.0036 | 0.2529±0.0040 | 0.5462±0.0140 | 0.41 |
| | Taylor | 1 | 27, 26, 25, 24, 28, 23, 29, 22 | 0.6088±0.0114 | 0.3288±0.0047 | 0.1660±0.0167 | 0.4318±0.0102 | 0.2790±0.0131 | 0.2310±0.0036 | 0.2534±0.0041 | 0.6093±0.0137 | 0.36 |
| | | 5 | 24, 26, 25, 28, 27, 23, 29, 22 | 0.6088±0.0114 | 0.3288±0.0047 | 0.1660±0.0167 | 0.4318±0.0102 | 0.2790±0.0131 | 0.2310±0.0036 | 0.2534±0.0041 | 0.6093±0.0137 | 0.36 |
| | | 10 | 24, 26, 25, 28, 27, 23, 29, 22 | 0.6088±0.0114 | 0.3288±0.0047 | 0.1660±0.0167 | 0.4318±0.0102 | 0.2790±0.0131 | 0.2310±0.0036 | 0.2534±0.0041 | 0.6093±0.0137 | 0.36 |
| | | 30 | 24, 23, 25, 26, 22, 27, 28, 20 | 0.7280±0.0104 | 0.4985±0.0050 | 0.2460±0.0193 | 0.6961±0.0094 | 0.4130±0.0144 | 0.6611±0.0038 | 0.4915±0.0046 | 0.7032±0.0128 | 0.55 |
| | | 50 | 24, 23, 25, 26, 22, 27, 28, 20 | 0.7280±0.0104 | 0.4985±0.0050 | 0.2460±0.0193 | 0.6961±0.0094 | 0.4130±0.0144 | 0.6611±0.0038 | 0.4915±0.0046 | 0.7032±0.0128 | 0.55 |

Table O: The effect of number of calibration samples on LLM layer pruning.

| Baseline | # Parameters (TTokens) | Benchmarks | | | | | | | | Avg Acc |
|---|---|---|---|---|---|---|---|---|---|---|
| | | PIQA | HellaSwag | OpenbookQA | ARC-e | ARC-c | MMLU | CMMLU | WinoGrande | |
| Vicuna-7B-v1.5 | 6.74B (370M) | 0.7720 | 0.5642 | 0.3300 | 0.7555 | 0.4326 | 0.4858 | 0.3518 | 0.6953 | 0.5484 |
| ChatGLM2-6B | 6.24B (1.4T) | 0.5403 | 0.2589 | 0.1420 | 0.2597 | 0.2005 | 0.2431 | 0.2537 | 0.5288 | 0.3034 |
| Baichuan2-7B | 7.51B (2.6T) | 0.7666 | 0.5363 | 0.3020 | 0.7475 | 0.4206 | 0.5024 | 0.5220 | 0.6819 | 0.5599 |
| Qwen1.5-7B | 7.72B (18T) | 0.7845 | 0.5785 | 0.3160 | 0.7125 | 0.4053 | 0.5967 | **0.7277** | 0.6575 | 0.5973 |
| LLaMA3-8B | 8.03B (15T+) | 0.7965 | 0.6014 | **0.3480** | 0.8005 | 0.4983 | 0.6212 | 0.4752 | 0.7332 | 0.6093 |
| Gemma2-7B | 8.54B (6T) | **0.8025** | **0.6039** | 0.3300 | 0.8110 | 0.5009 | 0.6143 | 0.4430 | **0.7435** | 0.6061 |
| Llama-3.1-8B-It | 8.03B (15T+) | 0.8003 | 0.5910 | 0.3380 | **0.8182** | **0.5179** | **0.6790** | 0.5552 | 0.7395 | **0.6299** |
| Llama-3-6.3B-Alpaca | 6.29B (12.74M) | 0.7388 | 0.5476 | 0.3160 | 0.7218 | 0.4394 | 0.6179 | 0.4497 | 0.6748 | 0.5633 |
| Llama-3.1-6.3B-It-Alpaca | 6.29B (12.74M) | 0.7383 | 0.5323 | 0.3080 | 0.7260 | 0.4684 | 0.6567 | 0.5515 | 0.6646 | 0.5807 |

Table P: Performance of our pruned models with respect to similarly-sized community models. "TTokens" denotes the training tokens. The best results are marked in **boldface**, and the sub-optimal ones are underlined.

| Model | Generated Sentences |
|-------|---------------------|
| Llama-3.1-6.3B-It-Alpaca | To be or not to be, that is the question that has been asked thousands of times throughout history. but have you ever wondered what it would be like to not exist? what would it feel like to not have consciousness or awareness of any kind? would it be like being asleep or a combed out hair? no, it would be something else entirely. it would be like not being here at all, like being a tree or a rock, a piece of furniture or a piece of trash. nothingness is not something we can really comprehend because we can't experience it. we can't even imagine it because we can't imagine. |
| | It fills me with such pride and joy knowing that I have been able to help shaped the lives of countless students over the years. I am proud of the numerous accomplishments and achievements of my students, and I am proud of the role I have played in shaping them into the people they are today. As a teacher, I believe it is my duty to provide an optimal learning environment that fosters growth and development of my students. I believe in making learning fun and engaging, and I am dedicated to pushing the bounds of innovation in the classroom. I am dedicated to ensuring that my students leave my classroom with the knowledge and skills they need ... |
| | What's great about the holiday season? Christmas lights, Christmas shows, Christmas presents, Christmas holiday traditions. But what's not great about the holiday season? crowds, stress, Santa Claus, Christmas holiday stress, Christmas holiday stressors. It's important to remember to do things that help you relax during the holiday season, such as taking time for yourself, engaging in relaxation techniques, practicing mindfulness, engaging in physical activity, practicing gratitude, practicing self-care, engaging in activities that bring you joy, and spending time with loved ones. These are all important components of stressors prevention during the holiday season. Here are some tips to help you. ... |
| Llama-3.1-8B-It | To be or not to be, that is the question. How does one decide whether a being is alive or not? The issue of what constitutes life has puzzled philosophers and scientists alike for centuries. A being is alive if it meets the criteria of life, which are: Metabolism: The ability to carry out basic functions such as respiration, photosynthesis, and nutrient uptake. Homeostasis: The ability to maintain a stable internal environment, despite changes in the external environment. Growth and Development: The ability to increase in size and complexity over time. Reproduction: The ability to produce offspring, either sexually or asex |
| | It fills me with such pride and joy to see my friends' passions ignite into flame and to watch them soar to great heights! It was an absolutely honour to have been able to share this experience with you and to bear witness to your hard work and dedication! It was an absolute pleasure to get to know you and share our experiences, but I know that this is just the beginning of an amazing journey! I feel so grateful to have you in my life, and I am constantly in awe of your love, kindness, and generosity! It's been such an incredible journey watching you discover and grow, and I feel so lucky to have... |
| | What's great about the holiday season? In addition to being able to spend time with loved ones and get some much-needed R&R, many people enjoy the idea of giving back to others. Whether it's volunteering, donating to charity, or participating in a Secret Santa gift exchange, the holiday season can be a time of kindness and generosity. But have you ever thought about how you might be able to combine your love of cooking and giving back this holiday season? If so, you might be interested in hosting a charity-themed potluck dinner or bake sale. Here are a few ideas to get you started: Host a potluck dinner to... |

Table Q: Generated Examples from the Llama-3.1-6.3B-It-Alpaca and Llama-3.1-8B-It.

