# OpenReview forum: "Reassessing Layer Pruning in LLMs: New Insights and Methods"
_ICLR.cc/2026/Conference — ICLR 2026 Poster_

### Official Review · Reviewer_PD7z · 2025-10-18

**Soundness:** 2
**Presentation:** 3
**Contribution:** 1
**Rating:** 2
**Confidence:** 4

**Summary:**

The paper relates to the pruning of LLM layers. The paper consists of three main parts:
1. Discussion of criteria for identifying prunable layers
2. Comparison between LoRA and partial fine-tuning methods for recovering accuracy after pruning
3. Theoretical analysis of gradient flow in the presence Pre-Layer Normalization, and how this affects layers by depth

The main observation in the paper is the relative unimportance of deep layers, and the fact that pruning the last layers is a more useful heuristic than other more elaborate importance estimators (c.f. Magnitude, Taylor, PPL, BI).
This claim is supported by Table 1, which shows superior results for the "reverse order" method, at a 20% pruning ratio, for Qwen1.5-7B, Llama-3.1-8B-It and Vicuna-7B-v1.5

A parallel finding is the fact that partial fine-tuning of the last one or two layers yields a greater accuracy recovery than full LoRA fine-tuning.
This claim is supported by Table 2.

In the last paragraph of the main body of the paper, the theoretical analysis of gradient flow and show that Pre-LN architectures inherently weaken the gradients and contributions of deeper layers due to the normalization step scaling them down.

**Strengths:**

The paper is nicely written, and very well laid out, making it an enjoyable read.

**Weaknesses:**

The paper focuses on depth pruning, however there is abundant evidence in the literature that layer-wise pruning is not as efficient as width pruning. For example, this claim is made in Muralidharan et al. (2024), which is cited in this paper.

In the main body of the paper, results are shown for a pruning ratio of 25%. We need to read the appendix to see results for 50% pruning ratio in Table G, and these results seem to contradict the main finding of the paper, since the "reverse order" method yields inferior results there. The PPL method appears to dominate at 50% pruning ratio, and the "random" method even wins the benchmark for Qwen1.5-7B, which raises questions about the relevance of the results.

The the LoRA vs. partial fine-tuning experiments, the study is limited to partial fine-tuning of the last few layers. Table 2 shows that fine-tuning the last three layers is better than fine-tuning the last two layers, which is better than fine-tuning the last layer. Thus, why stop at three layers? It would seem like if the trend follows, fine-tuning all layers would be optimal?

The theoretical analysis builds upon prior analyses of Pre-LN vs. Post-LN Transformers (e.g., Xiong et al., 2020; Liu et al., 2020). It's known that Pre-LN helps with training stability by damping gradients as depth increases, avoiding explosions near the output (which Post-LN can cause without warmups). However the theoretical analysis falls short of proving the optimality of fine-tuning just the last three layers.

I could not access any of the files behind the URL (https://anonymous.4open.science/r/Navigation-LLM-layer-pruning-DEB7/README.md) due to "The requested file is not found".

**Questions:**

"reverse order" wins the benchmark at 25% pruning ratio, but does not perform well at 50% pruning ratio, would you be able to do a comprehensive sweep of pruning ratios in order to collect more data points? For example, from 5% to 75%, by increments of 5%.

Can you repeat the experiments in Table 2 with partial-to-full fine-tuning, so we can see which setting is optimal in experimental results?

---

> ### Author Response · Authors · 2025-11-22
>
> Thank you for your detailed feedback. We address your questions below point by point:
>
> > The paper focuses on depth pruning, however there is abundant evidence in the literature that layer-wise pruning is not as efficient as width pruning. For example, this claim is made in Muralidharan et al. (2024), which is cited in this paper.
>
> Thank you for raising this concern. First, we want to point out that layer pruning is also an efficient, GPU-friendly method. As we described in our paper, layer pruning significantly reduces computational complexity and memory usage by directly removing model layers. To verify this, we conduct an experiment to test its efficiency. Specifically, we test the throughput of the pruned models obtained through our layer pruning and the width pruning method LLM-Pruner at the same pruning rate. The experimental results are shown in the table below. Our layer pruning method achieves a throughput of 30.92 tokens/s, significantly outperforming the width pruning method (25.76 tokens/s). This demonstrates that our method provides superior inference acceleration compared to width pruning. Besides, [1] also proves that layer pruning has a better inference acceleration effect than width pruning.
> | Method | Type |Throughput |
> | :--- | :--- |:--- |
> | LLM-Pruner | width pruning| 25.76  tokens/s |
> | Ours |layer pruning| 30.92 tokens/s |
>
> Furthermore, we have compared our layer pruning method with the structured width pruning method in the paper. In Section 5 (Table 4), our method significantly outperforms LLM-Pruner (a well-known width pruning method) on both Llama-3.1-8B-It and Llama-3-8B.
>
> > The results of the 50% pruning ratio in Table G.
>
>  Thank you for raising this concern. The observation in Table G is consistent with our theoretical analysis and empirical trends discussed in the main paper. Although reverse-order pruning does not always rank first at the 50% pruning ratio, its performance remains highly competitive: it ranks 2nd, 4th, and 1st out of 7 methods on Vicuna-7B-v1.5, Qwen1.5-7B, and Llama-3.1-8B-It, respectively. This deviation from the perfect ranking seen at the 25% pruning ratio is expected rather than contradictory.
>
> As established in Section 4.1, deeper layers tend to have smaller gradient contributions and are therefore more dispensable. Reverse-order pruning benefits specifically from removing these late layers. However, when the pruning ratio becomes as high as 50%, the pruning boundary moves significantly toward the shallow part of the network, where layers are inherently more important. In this regime, the advantage of exclusively pruning the deepest layers naturally diminishes, and performance differences among pruning criteria become smaller. This trend is clearly visualized in Figure A of the appendix, which shows a sharp performance drop once pruning reaches approximately the last 15–16 layers (around a 50% ratio). Thus, the 50% results do not contradict our main conclusion; instead, they reinforce the structural insight that reverse-order pruning is most effective when the pruning boundary remains in the region where deep layers dominate.
>
> Overall, the 50% experiments confirm the expected behavior: reverse-order pruning is consistently strong at moderate pruning ratios, and even at high pruning ratios it remains among the top-performing strategies.

---

> ### Author Response · Authors · 2025-11-22
>
> > The LoRA vs. partial fine-tuning experiments.
>
> Thank you for the question. We agree that extending partial fine-tuning beyond the last three layers is valuable. In the revision, we include experiments for more settings (e.g., last 4, 5, 6, 7, 8, 9, 10 layers) using Llama-3.1-8B-Instruct. As shown in the table below, as the number of fine-tuned layers increases, the average accuracy improves slightly, but the increase is very limited. However, during this process, GPU memory usage increases from 48.02GB to 69.99GB, and training time also increases from 7931.36s to 10316.53s. Therefore, taking into account both performance and computational cost, fine-tuning the last3 layer captures most of the performance recovery gains with lower training resource consumption.
>
> | Type | MMLU | CMMLU | PIQA | OpenbookQA | WinoGrande | HellaSwag | ARC-e | ARC-c | Avg Acc | GPU memory | Training time (2 epoch) |
> | :--- | :--- | :--- | :--- | :--- | :--- | :--- | :--- | :--- | :--- | :--- | :--- |
> | last3 | 0.6567 | 0.5515 | 0.7383 | 0.3080 | 0.6646 | 0.5323 | 0.7260 | 0.4684 | 0.5807 | 48.02G | 7931.36s |
> | last4 | 0.6487 | 0.5530 | 0.7405 | 0.2940 | 0.6685 | 0.5370 | 0.7285 | 0.4625 | 0.5791 | 49.50G | 8242.46s |
> | last6 | 0.6429 | 0.5545 | 0.7448 | 0.3080 | 0.6811 | 0.5391 | 0.7420 | 0.4659 | 0.5848 | 51.49G | 8572.35s |
> | last7 | 0.6414 | 0.5495 | 0.7476 | 0.3040 | 0.6898 | 0.5418 | 0.7462 | 0.4633 | 0.5855 | 55.18G | 8902.82s |
> | last7 | 0.6487 | 0.5530 | 0.7405 | 0.2940 | 0.6685 | 0.5370 | 0.7285 | 0.4625 | 0.5791 | 58.89G | 9289.69s |
> | last8 | 0.6445 | 0.5441 | 0.7497 | 0.2980 | 0.6859 | 0.5457 | 0.7572 | 0.4582 | 0.5854 | 62.59G | 9607.29s |
> | last9 | 0.6487 | 0.5432 | 0.7519 | 0.2960 | 0.6835 | 0.5475 | 0.7529 | 0.4642 | 0.5860 | 66.30G | 9975.59s |
> | last10 | 0.6576 | 0.5437 | 0.7470 | 0.3000 | 0.6748 | 0.5497 | 0.7521 | 0.4753 | 0.5875 | 69.99G | 10316.53s |
>
> > The theoretical analysis falls short of proving the optimality of fine-tuning just the last three layers.
>
> Thank you for pointing this out. We would like to clarify that our theoretical analysis does not aim to prove that fine-tuning exactly the last three layers is optimal. In fact, we do not claim that “three” is in any sense the theoretically best choice. Fine-tuning the last three layers is done with performance and computational costs in mind, as it can achieve most of the performance improvements with relatively low training resource consumption.
>
> The purpose of Table 2 is different. We aim to compare LoRA vs. full fine-tuning of the last k layers under similar parameter budgets, and the settings “k = 1, 2, 3” serve as representative examples. In the revised version, we will extend the experiments to larger values of k. As shown in the table above, as the number of fine-tuned layers increases, the average accuracy improves slightly, but the improvements become marginal.
>
> However, our theoretical analysis is relevant in determining which layers should be tuned first. As shown in Equation (15), model training or fine-tuning tends to induce larger changes in shallower layers, while deeper layers receive weaker updates. After pruning, the deepest remaining layers experience the largest distribution shift and are therefore the most critical to adapt. This supports the design choice of fine-tuning from the deepest layers upward (i.e., last-layer → last-two → last-three), even though the exact cutoff (three layers vs. more) is empirical rather than theoretically fixed.
>
> In summary, we do not claim that fine-tuning only three layers is theoretically optimal. Fine-tuning more layers is indeed better, but with diminishing returns. Our analysis in Section 4.2 supports the direction of fine-tuning (from deep to shallow), not a specific number of layers. The chosen settings are representative comparisons under comparable parameter budgets.
>
> > I could not access any of the files behind the URL (https://anonymous.4open.science/r/Navigation-LLM-layer-pruning-DEB7/README.md) due to "The requested file is not found".
>
>  Regarding your issue, we have re-checked. On our end, we try accessing the link and find that the files (including README.md) could be opened normally. We sincerely suggest you try accessing it again.

---

> > ### Comment · Reviewer_PD7z · 2025-11-24
> >
> > Thank you for posting this update, the answer addresses my concerns about the LoRA experiments.

---

> > > ### Comment · Reviewer_PD7z · 2025-11-24
> > >
> > > Comment https://openreview.net/forum?id=04Tfwy3LLC&noteId=duQCbF2eBR alleviates my concern about width vs layer pruning, as the improved throughput with layer pruning over width pruning is significant.
> > >
> > > In light of this and previous experimental results I am willing to revise my assessment of the paper.

---

> > > > ### Author Response · Authors · 2025-11-25
> > > >
> > > > Thank you for your active feedback and engagement during the rebuttal process. We appreciate your endorsement!

---

> ### Author Response · Authors · 2025-11-22
>
> > Do a comprehensive sweep of pruning ratios in order to collect more data points.
>
> To address your concerns and further demonstrate the effectiveness of our pruning method, we have added experiments at three different pruning rates: ~10%, ~20%, and ~35%. Specifically, we evaluate the performance of pruning models obtained through various pruning methods under different pruning rate settings. All baseline results are from [2]. Since this paper primarily tests datasets such as MMLU, PIQA, WinoGrande, ARC-e, and ARC-c, our comparisons also focus on these evaluation tasks to ensure that the comparisons with the baselines are fair. The experimental results are shown in the table below. At each pruning rate setting (10%, 20%, 35%), our method achieves the highest average accuracy, further demonstrating the effectiveness of our method.
> | Pruning Rate | Method | MMLU | PIQA | WinoGrande | ARC-e | ARC-c | Avg Acc |
> | :--- | :--- | :--- | :--- | :--- | :--- | :--- | :--- |
> | 10.86% | ShortGPT | 0.5800 | 0.7750 | 0.7020 | 0.7120 | 0.4740 | 0.6486 |
> | 10.86% | LaCo | 0.5880 | 0.7630 | 0.7230 | 0.7370 | 0.4890 | 0.6600 |
> | 10.16% | SliceGPT | 0.4390 | 0.7100 | 0.6720 | 0.6620 | 0.3940 | 0.5754 |
> | 10.86% | 2SSP | 0.5410 | 0.7970 | 0.7160 | 0.7160 | 0.4680 | 0.6476 |
> | 10% | COMPACT | 0.5960 | 0.7840 | 0.7370 | 0.7490 | 0.5030 | 0.6738 |
> | **12.50%** | **Ours** | **0.6785** | **0.7742** | **0.6993** | **0.7656** | **0.4949** | **0.6825** |
> | 19.02% | ShortGPT | 0.5860 | 0.7160 | 0.6840 | 0.5830 | 0.4220 | 0.5982 |
> | 19.02% | LaCo | 0.2410 | 0.7240 | 0.5530 | 0.5110 | 0.2920 | 0.4642 |
> | 20.12% | SliceGPT | 0.2450 | 0.6190 | 0.6180 | 0.4900 | 0.3030 | 0.4550 |
> | 19.99% | 2SSP | 0.3740 | 0.7680 | 0.6840 | 0.6180 | 0.3810 | 0.5650 |
> | 20% | COMPACT | 0.5070 | 0.7590 | 0.7010 | 0.6600 | 0.4280 | 0.6110 |
> | **21.88%** | **Ours** | **0.6378** | **0.7432** | **0.6654** | **0.7445** | **0.4744** | **0.6531** |
> | 35.31% | ShortGPT | 0.2320 | 0.5720 | 0.5910 | 0.3690 | 0.2970 | 0.4122 |
> | 35.31% | LaCo | 0.2310 | 0.5880 | 0.5310 | 0.3150 | 0.2730 | 0.3876 |
> | 35.16% | SliceGPT | 0.2300 | 0.5500 | 0.5430 | 0.3720 | 0.2350 | 0.3860 |
> | 34.77% | 2SSP | 0.2530 | 0.6870 | 0.5930 | 0.4430 | 0.2710 | 0.4494 |
> | 34.99% | COMPACT | 0.3590 | 0.7060 | 0.6330 | 0.4840 | 0.3080 | 0.4980 |
> | **34.38%** | **Ours** | **0.6650** | **0.7067** | **0.6772** | **0.6705** | **0.4198** | **0.6278** |
>
> Regarding your suggestion to extend the pruning rate to 75%, we would like to emphasize that currently, almost no research in the field of LLM layer pruning conducts experiments exceeding 50%. This is because excessive layer pruning can cause irreversible damage to model performance, leading to a collapse. Under such over-pruning, all layer pruning metrics tend to exhibit similarly low performance, diminishing the significance of comparisons between different methods. We believe that the supplementary experiments with 10%, 20%, and 35% pruning rates are sufficient to demonstrate the effectiveness of our method.

---

> ### Author Response · Authors · 2025-11-22
>
> > Can you repeat the experiments in Table 2 with partial-to-full fine-tuning, so we can see which setting is optimal in experimental results?
>
> Thank you for raising this concern. To solve your problem, we conduct an additional experiment to compare the performance of a pruned model trained from scratch with a pruned model that only fine-tuned the last 3 layers. Due to the extremely high cost of "training from scratch" on larger models and the limited number of GPUs, we choose TinyLlama-1.1B as our test model. Specifically, we prune the last four layers. Then, we compare two approaches: fine-tuning only the last three layers of the remaining layers and training from scratch. The experimental results are shown in the table below:
> | Model | Finetune Method | PIQA | HellaSwag | OpenbookQ | ARC-e | ARC-c | MMLU | CMMLU | WinoGrande | avg |
> | :--- |:--- | :--- | :--- | :--- | :--- | :--- | :--- | :--- | :--- | :--- |
> | TinyLlama-1.1B | **Last 3 Layers (Ours)** | 0.6480 | 0.3937 | 0.1940 | 0.4520 | 0.2705 | 0.2497 | 0.2479 | 0.5541 | **0.3762** |
> |TinyLlama-1.1B | **From Scratch** | 0.6670 | 0.3879 | 0.1880 | 0.4714 | 0.2491 | 0.2478 | 0.2497 | 0.5667 | **0.3784** |
>
> The experimental results show that fine-tuning the last 3 layers after pruning achieves an average score of 37.62, while training from scratch achieves an average score of 37.84. This experiment strongly demonstrates that our method can achieve performance almost equivalent to training a model of the same size from scratch with a much lower computational cost than training from scratch. This fully illustrates the efficiency and feasibility of our method in practice.
>
> Finally, we have updated the submission documents. Please see the updated PDF file for more information. Thank you for your detailed comments. Please don’t hesitate to reach out with any further questions. We would be happy to discuss further.
>
> [1] Yang, Yifei, et al. "LaCo: Large Language Model Pruning via Layer Collapse" Findings of the Association for Computational Linguistics: EMNLP 2024
>
> [2] Kwek, Eugene, and Wenpeng Yin. "COMPACT: Common-token Optimized Model Pruning Across Channels and Tokens." arXiv preprint arXiv:2509.06836 (2025).

---

### Official Review · Reviewer_Ncfm · 2025-10-27

**Soundness:** 3
**Presentation:** 3
**Contribution:** 3
**Rating:** 6
**Confidence:** 4

**Summary:**

This paper re-evaluates layer pruning methods for Large Language Models (LLMs), addressing whether complex metrics are needed to identify redundant layers and if LoRA is the optimal fine-tuning choice after pruning. Through extensive experiments across various metrics, LLMs, and fine-tuning methods, the paper reveals that a simple "backward pruning" (removing the last few layers directly) often outperforms more complex indicators. Furthermore, "partial layer fine-tuning" (tuning only the last few layers and the output layer) is found to be more effective and faster than LoRA for performance recovery. This paper provide a theoretical framework based on gradient flow to explain why deeper layers in Pre-LN Transformers contribute less, validating their approach. Pruned models based on these findings significantly surpass existing methods across benchmarks.

**Strengths:**

1.Comprehensive experimental design covering diverse pruning metrics, fine-tuning methods, and models.

2.The proposed "backward pruning + partial layer fine-tuning" strategy is simple yet effective.

3.Theoretical analysis using gradient flow provides a rationale for the method's efficacy.

4.Achieves significant performance gains across multiple models, outperforming other methods.

**Weaknesses:**

1.Inconsistent calibration datasets and data volumes were used for different pruning metrics, which could affect experimental fairness.

2.The performance of the pre-pruned models should be included in the results tables.

**Questions:**

1.Could you show the results of different pruning metrics without any subsequent training?

2.Have you compared pruning using other metrics (e.g., cosine similarity, perplexity) followed by fine-tuning only the layers immediately surrounding the pruned sections?

3.Recent work suggests deeper LLM layers are crucial for reasoning[1]. Does direct pruning of the final layers impact reasoning capabilities? It would be beneficial to evaluate this method on mathematical and code-related tasks to assess its performance in reasoning.

[1] Song, Xinyuan, et al. "Demystifying the Roles of LLM Layers in Retrieval, Knowledge, and Reasoning." arXiv preprint arXiv:2510.02091 (2025).

If the author's response addresses my questions, I will consider increasing my rating.

---

> ### Author Response · Authors · 2025-11-22
>
> Thank you for your thorough review, and for providing insightful feedback and support. We address your questions below point by point:
>
> > Inconsistent calibration datasets and data volumes were used for different pruning metrics, which could affect experimental fairness.
>
> Thank you for your interest in experimental fairness. Fairness is also a goal we strive for when designing experiments. We would like to clarify three points:
>
> - Our method (reverse pruning) does not require calibration data: We want to emphasize first that reverse pruning is a **completely data-free** method. This method is so simple that it involves directly cutting off the last (deepest) layers of the model.
> - The amount of calibration data does affect the performance of other pruning methods: In fact, this is an issue we explored in depth in Table N. Our experiments  show that for data-driven metrics like BI and Taylor, the number of calibration samples (1, 5, 10, 30, 50) significantly impacts their pruning performance.
> - Our baseline reproduction guarantees fairness: Finally, to ensure that our comparisons with existing methods are fair, we want to clarify that in all experiments reproduced by us, we used the exact same calibration datasets and data volumes.
>
> > The performance of the pre-pruned models should be included in the results tables.
>
>  Thank you for your valuable suggestions. We have added benchmark results for pre-pruned models to the revised version.
>
> > Could you show the results of different pruning metrics without any subsequent training?
>
> Thank you for your insightful review. In our initial submission, we have compared the performance of different pruning metrics (SLEB, a well-known retraining-free pruning method) without any subsequent training in Table F of the Appendix. To further illustrate the effectiveness of our method and provide a more comprehensive comparison, we add a structured pruning method, LLM-Pruner. Specifically, we apply SLEB, LLM-Pruner, and reverse-order algorithms to prune the Llama-3.1-8B-It and Llama-3-8B models, respectively, and then evaluate their zero-shot performance directly on benchmarks without any fine-tuning or training. As shown in the table below, even without any fine-tuning, our simple reverse-order pruning significantly outperforms SLEB and LLM-Pruner in terms of average accuracy on Llama-3.1-8B-It and Llama-3-8B. This further demonstrates the effectiveness of our method, making it highly competitive even as a pure, retraining-free compression technique.
>
> | Model | Method | PIQA | HellaSwag | OpenbookQA | ARC-e | ARC-c | MMLU | CMMLU | WinoGrande | Avg Acc |
> | :--- | :--- | :--- | :--- | :--- | :--- | :--- | :--- | :--- | :--- | :--- |
> | Llama-3.1-8B-It | SLEB (w/o fine-tuning) | 0.7252 | 0.4415 | 0.2380 | 0.6423 | 0.3166 | 0.3396 | 0.2756 | 0.5888 | 0.4192 |
> | Llama-3.1-8B-It | LLM-Pruner (w/o fine-tuning) | 0.7356 | 0.4518 | 0.2760 | 0.6713 | 0.3473 | 0.3413 | 0.2704 | 0.6559 | 0.4687 |
> | Llama-3.1-8B-It | Reverse-order (w/o fine-tuning) | 0.7002 | 0.4021 | 0.2920 | 0.6178 | 0.3993 | 0.6346 | 0.5458 | 0.6251 | **0.5271** |
> | Llama-3-8B | SLEB (w/o fine-tuning) | 0.7111 | 0.4401 | 0.2280 | 0.6014 | 0.2807 | 0.2674 | 0.2502 | 0.5683 | 0.3689 |
> | Llama-3-8B | LLM-Pruner (w/o fine-tuning) | 0.7220 | 0.4578 | 0.2840 | 0.6338 | 0.3319 | 0.2700 | 0.2612 | 0.6125 | 0.4467 |
> | Llama-3-8B | Reverse-order (w/o fine-tuning) | 0.6921 | 0.4035 | 0.3040 | 0.6014 | 0.3720 | 0.5603 | 0.4216 | 0.5975 | **0.4940** |
>
> > Have you compared pruning using other metrics (e.g., cosine similarity, perplexity) followed by fine-tuning only the layers immediately surrounding the pruned sections?
>
> Thank you for your insightful suggestion. You are absolutely right. After pruning with other metrics, fine-tuning with our proposed partial-layer fine-tuning can further verify the effectiveness and robustness of our partial-layer fine-tuning strategy. Actually, we have already conducted this experiment in Appendix Table H of the initial manuscript (Table L of the revised version). Experimental results show that after using Taylor pruning, the average accuracy using traditional LoRA for fine-tuning is only 0.4796. However, when we switch to our proposed partial-layer fine-tuning on the same pruned model, the average accuracy significantly improves to 0.5701. This experiment fully demonstrates that our partial-layer fine-tuning strategy is not only applicable to reverse-order pruning, but is also a universally effective performance recovery method. Thank you again for raising this important question, which gives us the opportunity to emphasize the robustness of our fine-tuning method.

---

> ### Author Response · Authors · 2025-11-22
>
> > Recent work suggests deeper LLM layers are crucial for reasoning. Does direct pruning of the final layers impact reasoning capabilities? It would be beneficial to evaluate this method on mathematical and code-related tasks to assess its performance in reasoning.
>
>  Thank you for your suggestion. Song et al.[2] find that deeper layers are crucial for reasoning and that performance can be recovered through distillation, which aligns with our experimental results. The performance recovery observes through distillation is similar to the effectiveness of our proposed partial-layer fine-tuning, which selectively retrains the remaining last few layers. We have integrated this paper into our revised manuscript to strengthen the connection between our empirical findings and existing research.
>
> Furthermore, we do lack testing for the complex reasoning capabilities (e.g., mathematics and coding) of the pruned model. To address your concerns, we conduct additional experiments. Specifically, we evaluate the performance of pruned models obtained by various pruning methods on GSM8K using Llama-3.1-8B. All baseline results are reported from [1]. The experimental results are shown in the table below. Despite our method having a higher pruning rate, our model accuracy is significantly higher than all baseline methods, reaching 29.19%. This further demonstrates the effectiveness of our proposed method on mathematical tasks.
> | Method | ShortGPT | LaCo | SliceGPT | 2SSP[3] | COMPACT[1] | Ours |
> | :--- | :--- | :--- | :--- | :--- | :--- | :--- |
> | Pruning Rate | 19.02% | 19.02% | 20.12% | 19.99% | 20.00% | 25.00% |
> | Acc (%) | 0.60 | 0.40 | 0.00 | 4.30 | 10.80 | **29.19** |
>
> Besides, we evaluate our method on code-related tasks. Specifically, the experiments use CodeQwen1.5-7B-Chat model with a pruning rate of 20%, and are tested on HumanEval. The experimental results are shown in the table below. Our method achieves the highest Pass@1 at 48.72, significantly exceeding ShortGPT (42.68) and LLM-Pruner (15.85).
> | Method | ShortGPT | UIDL[4] | Linearity[5] | SLEB | LLM-Pruner | Ours |
> | :--- | :--- | :--- | :--- | :--- | :--- | :--- |
> | Acc (Pass@1) | 42.68 | 0.00 | 0.00 | 20.73 | 15.85 | **48.72** |
>
> Finally, the superior performance in mathematical and code benchmarks further validates the effectiveness and generality of our method.
>
> We have updated the submission documents. Please see the updated PDF file for more information. Thank you for your detailed comments. We hope that our additional experiments and clarifications have addressed your concerns. Please don’t hesitate to reach out with any further questions. We would be happy to discuss further.
>
> [1] Kwek, Eugene, and Wenpeng Yin. "COMPACT: Common-token Optimized Model Pruning Across Channels and Tokens." arXiv preprint arXiv:2509.06836 (2025).
>
> [2] Song, Xinyuan, et al. "Demystifying the Roles of LLM Layers in Retrieval, Knowledge, and Reasoning." arXiv preprint arXiv:2510.02091 (2025).
>
> [3] Sandri, Fabrizio, Elia Cunegatti, and Giovanni Iacca. "2ssp: A two-stage framework for structured pruning of llms." arXiv preprint arXiv:2501.17771 (2025).
>
> [4]Gromov, Andrey, et al. "The unreasonable ineffectiveness of the deeper layers." arXiv preprint arXiv:2403.17887 (2024).
>
> [5] Razzhigaev, Anton, et al. "Your transformer is secretly linear." arXiv preprint arXiv:2405.12250 (2024).

---

> ### Comment · Reviewer_Ncfm · 2025-11-23
>
> Thank you for the detailed response. Regarding question 2 that I mentioned earlier, what I meant was: if cosine similarity is used for pruning consecutive layers—for instance, if layers 20 to 28 are pruned—then we could freeze all other layers and only train a few layers before layer 20 and after layer 28, while keeping the number of trainable layers the same as in your method. I would like to know how the results of such an approach would compare with those of your method.

---

> > ### Author Response · Authors · 2025-11-25
> >
> > Thank you for your feedback. We appreciate the opportunity to address your concerns and improve the quality of our work. Here, we provide additional analysis and clarification to address your comments.
> >
> > First, we use cosine similarity to identify redundant layers according to your requirements. We find that in the Llama-3.1-8B-It model, the 8 unimportant layers selected by cosine similarity are 2, 3, 4, 5, 6, 7, 10, and 13, which are discontinuous layers.To satisfy your assumptions and make a fair comparison, we forcibly remove the 8 consecutive layers from 20 to 27. Then we only fine-tune the layers before and after the deleted contiguous blocks. To ensure that the number of trainable parameters is consistent with our method, we select the last two layers of the remaining layers for fine-tuning. As shown in table below, our method achieves significantly higher average accuracy than fine-tuning the layers before and after the deleted contiguous blocks. We believe that this experiment further demonstrates the effectiveness of our method. If you have any further questions, we would be happy to answer them for you.
> >
> > | Method | MMLU | CMMLU | PIQA | OpenbookQA | WinoGrande | HellaSwag | ARC-e | ARC-c | Avg Acc |
> > | :--- | :--- | :--- | :--- | :--- | :--- | :--- | :--- | :--- | :--- |
> > | Two layers before and after the deleted block | 0.6453 | 0.5186 | 0.7195 | 0.2810 | 0.7026 | 0.5333 | 0.7189 | 0.4408 | 0.5663 |
> > | Last two layers (ours) | 0.6568 | 0.5543 | 0.7361 | 0.3080 | 0.6567 | 0.5204 | 0.7151 | 0.4633 | 0.5766 |

---

> > > ### Comment · Reviewer_Ncfm · 2025-11-25
> > >
> > > Thank you for the additional experiments. The results align with my expectations. Although reverse-order pruning with fine-tuning of the last few layers outperforms the cosine similarity-based approach (which fine-tunes both earlier and later layers), the difference in performance is not significant. I have raised my score to 8 to support the paper's acceptance.

---

> > > > ### Author Response · Authors · 2025-11-25
> > > >
> > > > We are deeply grateful for your extensive involvement and your ultimately generous support. Your decision is a tremendous encouragement to us.

---

### Official Review · Reviewer_mE3b · 2025-10-30

**Soundness:** 2
**Presentation:** 3
**Contribution:** 2
**Rating:** 4
**Confidence:** 4

**Summary:**

This paper re-evaluates layer pruning for Pre-LN LLMs and shows that a simple strategy that prunes layers in reverse order and then fine-tune only the LM head plus the last 1-3 layers consistently matches or even outperforms more complicated pruning methods on a few standard benchmarks (PIQA, HellaSwag, WinoGrande, ARC-e/c, OBQA, MMLU, CMMLU). The empirical study is broad (several LLaMA and Qwen-style models) and scales up to LLaMA-3-70B. The authors give gradient-flow explanation for why deeper layers in Pre-LN are matter less, and they also find that this approach can beat the usual "prune + LoRA" recovery. This makes the paper especially useful for users who just want a reliable pruning recipe without complex per-layer scoring.

**Strengths:**

- Clear recipe to prune layers in reverse order and fine-tune only the LM head alongside the last 1–3 layers.
- Reasonable empirical baking, tested on several LLaMA-3 and Qwen-style models at several pruning ratios, and several standard benchmarks, and it still works at 70B scale.
- Practical impact, simple post-pruning FT outperforms the common "prune + LoRA" setup.
- Plausible architectural explanation, the Pre-LN gradient-flow analysis motivates why late layers are safer to drop.

**Weaknesses:**

- They don’t evaluate on generation or reasoning datasets (e.g. GSM8K), so the conclusions are validated only on specific LM-harness-style multiple-choice tasks.
- Prior work shows that layer importance depends on the nature of the task. Without generation tasks, the paper assumes task-invariance of the "prune-from-the-top" rule. Later layers tend to be more critical for perplexity, so pruning them first might hurt exactly the tasks they didn’t test.
- As a result, the current recipe is a strong default for classification-style LLM evals, but its generality to generation remains unproven.

**Questions:**

- Can you add some generation/reasoning benchmarks (e.g. GSM8K) to verify that reverse-order pruning still holds outside multiple choice tasks?
- Please also report the perplexity on some perplexity based data sets e.g. wikitext to see how that varies across varies different techniques.
- Do the results apply for different model architectures as well? (it would be interesting to see the results on some models mixture of experts models e.g. Mixtral 8×7B)

I'd be happy to increase my score if these experiments are included!

---

> ### Author Response · Authors · 2025-11-22
>
> We greatly appreciate your positive feedback and constructive suggestions, which have been instrumental in improving the quality of our work.
>
> > They don’t evaluate on generation or reasoning datasets (e.g. GSM8K).
>
> Thank you for your suggestion. We choose these benchmark datasets primarily to follow established criteria in the pruning literature (e.g.,[1]) to ensure a fair comparison with previous state-of-the-art methods. Following your suggestion, we have added a comparison of pruning performance on GSM8K. Specifically, we evaluate the accuracy of the pruned Llama-3.1-8Bs obtained by various pruning methods on GSM8K. All baseline results are from [2]. The experimental results are shown in the table below. Despite our method having a higher pruning rate, our model achieves a significantly higher accuracy than all baseline methods, reaching 29.19%. This further demonstrates the versatility and effectiveness of our proposed method.
> | Method | ShortGPT | LaCo | SliceGPT | 2SSP[3] | COMPACT[2] | Ours |
> | :--- | :--- | :--- | :--- | :--- | :--- | :--- |
> | Pruning Rate | 19.02% | 19.02% | 20.12% | 19.99% | 20.00% | 25.00% |
> | Acc (%) | 0.60 | 0.40 | 0.00 | 4.30 | 10.80 | **29.19** |
>
> > Report the perplexity on some perplexity based data sets e.g. wikitext.
>
> Thanks for your suggestion. We agree that perplexity is an important metric for evaluating the language modeling capabilities of pruned models. To answer your question, we evaluate the performance of various pruned models (20% pruning rate, Llama-3-8B) on WikiText-2. Experimental results show that our method achieves the lowest perplexity value of 19.16, significantly outperforming all other pruning methods. This further demonstrates that our proposed method can maximally preserve the general generative capabilities of the pruned model.
> | Method | Wanda | LLM-Pruner | ShortGPT | OWL | GISP | Ours |
> | :--- | :--- | :--- | :--- | :--- | :--- | :--- |
> | Perplexity | 29.92 | 23.21 | 118.62 | 29.49 | 24.18 | **19.16** |
>
> > Do the results apply for different model architectures as well? (it would be interesting to see the results on some models mixture of experts models e.g. Mixtral 8×7B)
>
> Thank you for your interest in the effectiveness of our pruning algorithm across different model architectures, especially Mixture-of-Experts models like Mixtral 8x7B. To answer your question, we conduct supplementary experiments on the Mixtral 8x7B model. Specifically, we prune the last 5 blocks of the Mixtral 8x7B model and compare it with a MoE pruning baseline UMD [4] at the same pruning rate. Experimental results show that our method achieves an average accuracy of 0.6673 on the MoE architecture, which is higher than UMD's 0.6558. This further demonstrates that our method has good architectural generality: even in complex MoE models, our simple "reverse pruning" strategy can still outperform existing MoE pruning baselines and achieve better performance.
> | Method | MMLU | PIQA | OpenbookQA | WinoGrande | HellaSwag | ARC-c | Avg Acc |
> | :--- | :--- | :--- | :--- | :--- | :--- | :--- | :--- |
> | UMD | 0.6790 | 0.7930 | 0.4200 | 0.7430 | 0.7870 | 0.5130 | 0.6558 |
> | Ours | 0.6957 | 0.7890 | 0.4321 | 0.7674 | 0.7981 | 0.5216 | **0.6673** |
>
> In addition, we have updated the submission documents. Please see the updated PDF file for more information. We appreciate your valuable feedback and welcome any further questions or discussions. Please let us know if there are other questions; we'd be happy to address them.
>
> [1] Ma, Xinyin, Gongfan Fang, and Xinchao Wang. "Llm-pruner: On the structural pruning of large language models." Advances in neural information processing systems 36 (2023): 21702-21720.
>
> [2] Kwek, Eugene, and Wenpeng Yin. "COMPACT: Common-token Optimized Model Pruning Across Channels and Tokens." arXiv preprint arXiv:2509.06836 (2025).
>
> [3] Sandri, Fabrizio, Elia Cunegatti, and Giovanni Iacca. "2ssp: A two-stage framework for structured pruning of llms." arXiv preprint arXiv:2501.17771 (2025).
>
> [4] He, Shwai, et al. "Towards Efficient Mixture of Experts: A Holistic Study of Compression Techniques." Transactions on Machine Learning Research.

---

> > ### Comment · Reviewer_mE3b · 2025-11-26
> > **Reviewer Response**
> >
> > Thank you for you answer, it answers my questions and I've revised my score accordingly.

---

> > > ### Author Response · Authors · 2025-11-26
> > >
> > > We sincerely appreciate your encouraging comment and valuable feedbacks.

---

### Official Review · Reviewer_tCfY · 2025-11-01

**Soundness:** 4
**Presentation:** 3
**Contribution:** 2
**Rating:** 8
**Confidence:** 3

**Summary:**

The paper is about empirical benchmarking and methodological clarification for layer pruning.

Benchmarks 7 layer-selection metrics and 6 fine-tuning methods across Vicuna-7B, Qwen-7B, and Llama-3.x models.

Finds that reverse-order pruning (dropping last layers) consistently outperforms complex importance metrics.

Shows partial-layer fine-tuning (LM head + last 1–3 layers) surpasses LoRA/QLoRA for accuracy and training cost.

Extends tests to Llama-3-70B.

Reports 2-19 pp improvement over prior layer-pruning baselines.

Adds a gradient-flow derivation explaining why deep layers matter less.

Notes that iterative prune–tune cycles provide no benefit over one-shot pruning.

**Strengths:**

Comprehensive and reproducible experimental design.

Honest ablations revealing when complexity adds no value.

Simple, clearly-defined recipe that practitioners can reproduce in hours.

Really primarily illustrates a weakness in all the other papers on layer pruning: they ought to have used final layer pruning as the obvious control experiment and have failed to do so. Providing this missing baseline is probably important within the narrow domain of layer pruning.

Experimentally verifies a fact that is part of the design of LLM architectures and their understanding as unrolling, and has also been examined theoretically and by other experimental methods before.

**Weaknesses:**

Scope: confined to layer pruning; ignores dominant GPU-friendly methods (structured width pruning, 2:4 sparsity, quantization).

Novelty: theoretical component re-derives known results; empirical finding is mainly that others’ metrics fail.

Practical relevance: minimal for most users in practice. For people training from scratch, incremental deepening is probably preferable. For people trying to squeeze a large model into a slightly smaller GPU, quantization and GPU-friendly sparsity are probably preferable even if smaller models aren't just available. The primary use case is where a user has an unusual model, cannot control training, but quickly wants to squeeze it into an existing GPU with somewhat more limited space.

The paper really ought to have compared final layer pruning with models of the same final size trained from scratch, since they are architecturally identical. This would indicate whether final layer pruning could be a useful shortcut for generating a simple multi-depth collection of models

**Questions:**

Ought to address:

- Clarify that the theoretical contribution is an application of prior analyses, not new theory.
- Discuss why reverse-order pruning should be a baseline control for future pruning papers.

I think the following is really future work:

- Benchmark versus quantized models at equal memory budgets.
- Investigate interaction of quantization and depth--does aggressive quantization change which layers are dispensable?
- Compare against GPU-usable sparsity (2:4) and width pruning for completeness.
- Compare with models trained from scratch at the final depth, as well as models incrementally grown.

---

> ### Author Response · Authors · 2025-11-22
>
> Thank you for your valuable time and thorough review of our paper. We are honored to receive your "accept" rating. Your comments are insightful and highly constructive, and crucial to improving our work. We have carefully read all your comments and have responded and explained accordingly.
>
> > Scope: confined to layer pruning; ignores dominant GPU-friendly methods (structured width pruning, 2:4 sparsity, quantization).
>
> Thank you very much for your pursuit of the core value of GPU-friendly methods. We fully agree that when deploying LLMs in resource-constrained environments, model efficiency and inference acceleration are primary considerations, which is also the reason why we systematically evaluate layer pruning. We want to point out that layer pruning is also an efficient, GPU-friendly method. As we described in our paper, layer pruning significantly reduces computational complexity and memory usage by directly removing model layers. To demonstrate this, we conduct a series of experiments to test its GPU-friendly performance. Specifically, we test the throughput of the pruned models obtained by our layer pruning method, channel pruning method (i.e., structured width pruning, LLM-Pruner), and weight pruning method (i.e., 2:4 sparsity, Wanda) under the same pruning rate. The experimental results are shown in the table below, where (our) layer pruning method demonstrates excellent throughput. Besides, [1] also demonstrates that layer pruning has the best inference acceleration effect.
>
> | Method | Type |Throughput |
> | :--- | :--- |:--- |
> | Wanda [1] | weight pruning |26.71 tokens/s |
> | LLM-Pruner | channel pruning| 25.76 tokens/s |
> | Ours |layer pruning| 30.92  tokens/s |
>
> Furthermore, we have already compared our layer pruning method with structured width pruning and weight pruning methods in our paper. In Section 5 (Table 4) of our paper, we include LLM-Pruner (a well-known width pruning method) and SliceGPT (a weight pruning method). Our method achieves state-of-the-art performance in these comparisons.
>
> Finally, regarding quantization, our layer pruning method is orthogonal to quantization, and the two can complement each other. Our method reduces computational complexity by decreasing depth, while quantization reduces memory usage by decreasing parameter precision. To demonstrate our claim, we further quantize the model obtained from our pruning method using GPTQ. The experimental results are shown in the table below. Our pruned model has a memory footprint of 11988.40 MB and an average accuracy of 0.5814. After applying GPTQ for quantization, the model's memory footprint drops dramatically to 1326.93 MB, achieving a reduction of over 89%. While this extreme compression results in the expected performance loss, the significant reduction in memory footprint demonstrates the immense potential of layer pruning and quantization as orthogonal techniques in practical deployments.
>
> | Method      | Memory       | MMLU   | CMMLU  | PIQA   | OpenbookQA | WinoGrande | HellaSwag | ARC-e  | ARC-c  | Avg Acc |
> |-------------|--------------|--------|--------|--------|-------------|-------------|------------|--------|--------|---------|
> | Ours        | 11988.40 MB  | 0.6567 | 0.5515 | 0.7383 | 0.3080      | 0.6646      | 0.5323     | 0.7260 | 0.4684 | 0.5807  |
> | Ours+GPTQ   | 1326.93 MB   | 0.6093 | 0.4571 | 0.7193 | 0.2860      | 0.6267      | 0.5090     | 0.6822 | 0.3985 | 0.5360  |
>
> > Novelty: theoretical component re-derives known results; empirical finding is mainly that others’ metrics fail.
>
> We believe that our core contribution is not merely the "failure of other metrics," but a counterintuitive conclusion: the simplest "reverse-order" method consistently outperforms a variety of complex and computationally expensive metrics. Furthermore, we find that fine-tuning only the last few layers and lm_head outperforms commonly used LLM pruning performance recovery methods such as LoRA, which represents a paradigm shift in the field of LLM pruning optimization. Our theoretical derivation aims to support our two core empirical findings: 1. Why pruning the last few layers is reasonable (because the gradient contribution of deeper layers is weakened in Pre-LN transformers) and 2. Why fine-tuning the last remaining layers is effective (because the last remaining layers after pruning require the greatest fitness).

---

> ### Author Response · Authors · 2025-11-22
>
> > Practical relevance: minimal for most users in practice. For people training from scratch, incremental deepening is probably preferable. For people trying to squeeze a large model into a slightly smaller GPU, quantization and GPU-friendly sparsity are probably preferable even if smaller models aren't just available. The primary use case is where a user has an unusual model, cannot control training, but quickly wants to squeeze it into an existing GPU with somewhat more limited space.
>
> Thank you very much for your in-depth consideration of the practical value of our pruning method. We would like to first clarify that, as mentioned in our previous reply, our layer pruning method is also an efficient GPU-friendly method. To demonstrate this, we specifically test throughput, as it best represents actual inference speed. Experiments show that our method significantly accelerates inference, outperforming representative weight pruning and channel pruning methods. Furthermore, our method is orthogonal to quantization, and the two can be used in combination. Our method reduces computational complexity by decreasing depth, while quantization reduces memory footprint by decreasing parameter precision. Finally, our method  aims to compress an existing large pre-trained model into a smaller, faster model for rapid deployment in resource-constrained environments. We argue that in the real world, most users lack control over training from scratch (due to high training costs), making this scenario of "compressing existing large models" both common and highly practical.
>
> > The paper really ought to have compared final layer pruning with models of the same final size trained from scratch, since they are architecturally identical. This would indicate whether final layer pruning could be a useful shortcut for generating a simple multi-depth collection of models.
>
> Thank you for your insightful suggestions. We strongly agree that comparing the pruned model to a model trained from scratch on the same size is the gold standard for evaluating the effectiveness of our pruning method. To this end, we conduct an additional experiment. Due to the extremely high cost of "training from scratch" on larger models and the limited number of GPUs, we choose TinyLlama-1.1B as our test model. Specifically, we prune the last four layers. Then, we compare two approaches: fine-tuning only the last three layers of the remaining layers and training from scratch. The experimental results are shown in the table below:
> | Model | Finetune Method | PIQA | HellaSwag | OpenbookQ | ARC-e | ARC-c | MMLU | CMMLU | WinoGrande | Avg Acc |
> | :--- |:--- | :--- | :--- | :--- | :--- | :--- | :--- | :--- | :--- | :--- |
> | TinyLlama-1.1B | **Last 3 Layers (Ours)** | 0.6480 | 0.3937 | 0.1940 | 0.4520 | 0.2705 | 0.2497 | 0.2479 | 0.5541 | 0.3762 |
> |TinyLlama-1.1B | **From Scratch** | 0.6670 | 0.3879 | 0.1880 | 0.4714 | 0.2491 | 0.2478 | 0.2497 | 0.5667 | 0.3784 |
>
> The experimental results show that our proposed method (fine-tuning the last 3 layers after pruning) achieves an average score of 37.62, while the baseline model trained from scratch achieves an average score of 37.84. This experiment strongly demonstrates that our method can achieve performance almost equivalent to training a model of the same size from scratch with a much lower computational cost than training from scratch. This fully illustrates the efficiency and feasibility of our method in practice.
>
> > Clarify that the theoretical contribution is an application of prior analyses, not new theory.
>
> As you pointed out, we apply previous analyses, particularly those concerning gradient flow in Pre-LN Transformers, to provide a solid theoretical explanation for our empirical findings. Specifically, we use these analyses to clarify why reverse pruning (removing the last few layers) and fine-tuning the last remaining layers are effective in our experiments. We will add explicit clarification in the final revised version, clearly positioning our theoretical contribution as a "successful application of previous analyses."

---

> ### Author Response · Authors · 2025-11-22
>
> > Discuss why reverse-order pruning should be a baseline control for future pruning papers.
>
> Thank you for your insightful and constructive suggestions. We will explain why reverse pruning should be used as the baseline for future pruning papers from the following three aspects.
> - Simplicity: Reverse pruning (i.e., pruning the last few layers) is the simplest and most direct pruning strategy. It does not require complex calculations like Taylor, PPL, or BI, nor does it require any calibration datasets; the computational cost is almost zero.
> - Effectiveness: Despite its simplicity, our empirical results (as shown in Tables 1, 4, and 5) consistently demonstrate its superior performance and consistently outperform many complex pruning metrics.
> - Theoretical Basis: The effectiveness of our pruning method is not accidental, but can be theoretically proven (Section 4). For the Pre-LN Transformer architecture (the mainstream architecture of LLM today), due to the weakening of gradient flow, deeper layers (i.e., the last few layers) do indeed contribute less to the overall performance of the model.
>
> Therefore, we believe that if a newly proposed, more complex pruning metric cannot even outperform the zero-cost baseline of reverse pruning, its practical value is highly questionable. Using it as the standard baseline can effectively prevent the community from pursuing unnecessary complexity.
>
> > Benchmark versus quantized models at equal memory budgets.
>
> As we mentioned in our previous response, pruning and quantization are two complementary and orthogonal compression techniques that can be used in combination. Our layer pruning focuses on reducing computational complexity and latency by decreasing model depth, while quantization focuses on reducing memory footprint by decreasing parameter precision. To prove this point, we use GPTQ to quantize our already pruned model. As shown in the table below, our basic pruned model uses 11988.40 MB of memory, with an average score of 0.5814. After we apply GPTQ quantization, the memory used drops quickly to only 1326.93 MB. This means we achieve a very large memory reduction of over 89%. This extreme saving causes an expected, small drop in performance, with the average score moving from 0.5814 down to 0.5360. However, because the memory saving is so massive, this shows the strong potential of using layer pruning and quantization together for deployment on devices with limited resources.
>
> | Method      | Memory       | MMLU   | CMMLU  | PIQA   | OpenbookQA | WinoGrande | HellaSwag | ARC-e  | ARC-c  | Avg Acc |
> |-------------|--------------|--------|--------|--------|-------------|-------------|------------|--------|--------|---------|
> | Ours        | 11988.40 MB  | 0.6567 | 0.5515 | 0.7383 | 0.3080      | 0.6646      | 0.5323     | 0.7260 | 0.4684 | 0.5807  |
> | Ours+GPTQ   | 1326.93 MB   | 0.6093 | 0.4571 | 0.7193 | 0.2860      | 0.6267      | 0.5090     | 0.6822 | 0.3985 | 0.5360  |
>
> > Investigate interaction of quantization and depth--does aggressive quantization change which layers are dispensable?
>
>  Thank you for raising such a profound and novel question. This is a core question about the interaction between model pruning and quantization.
>
> We have verified the feasibility of pruning followed by quantization: In our previous response, we demonstrate the effectiveness of applying quantization to the pruned model. Experiments show that this order drastically reduces memory usage from 11988.40 MB to 1326.93 MB while achieving an acceptable performance degradation, demonstrating the complementarity of the two methods.
> Your question about quantizing first and then evaluating its impact on layer importance ranking is a valuable approach. It suggests that quantization might alter gradient flow or activation distribution, thus making previously unimportant layers crucial. We strongly agree with your idea and will take this approach as a key direction for our future work, which will be elaborated upon in the future work section.
>
> We thank you again for your profound insights and for pointing us in the right direction for future work.

---

> ### Author Response · Authors · 2025-11-22
>
> > Compare against GPU-usable sparsity (2:4) and width pruning for completeness.
>
> We appreciate the reviewers' suggestions. We fully agree that comparing layer pruning with width pruning and GPU-usable sparsity is crucial. We would like to clarify that we have already made these comparisons in the paper. In Table 4, we compare our method with these state-of-the-art pruning methods, where LLM-Pruner is a width pruning method and SliceGPT represents a GPU-usable sparsity method. Our method shows advantages over all these methods, demonstrating the effectiveness of our layer pruning strategy in practice.
>
> > Compare with models trained from scratch at the final depth, as well as models incrementally grown.
>
> We greatly appreciate your idea of ​​comparing our method with these two baselines. In the supplementary experiments above, we have demonstrated that our method (reverse pruning + fine-tuning the last 3 layers) performs comparably to models of the same size trained from scratch, but with significantly lower training costs. Regarding the comparison of our method with incremental growth models, we believe this is a novel and valuable research direction. However, due to our limited computational resources, we are unable to conduct such large-scale and time-consuming incremental model training experiments. Therefore, we have listed this as one of the priorities for future work, hoping to fill this gap in future research. Thank you again for your valuable suggestion.
>
>
> Meanwhile, we have updated the submission documents. We do think these additions strengthen our work. Please take a look at the updated PDF to find out more. If you have any additional questions or concerns that we can clarify or address, we would be happy to provide further information to ensure all aspects of our work are clear. Thank you once again for your valuable time and effort in reviewing our submission!
>
> [1] Yang, Yifei, et al. "LaCo: Large Language Model Pruning via Layer Collapse" Findings of the Association for Computational Linguistics: EMNLP 2024
>
> [2] Sun, Mingjie, et al. "A Simple and Effective Pruning Approach for Large Language Models." ICLR.

---

### Author Response · Authors · 2025-12-02
**Final Rebuttal Summary to the Area Chair**

Dear AC and Reviewers, we sincerely appreciate your review of our work and your responsible management of the discussion process. To facilitate a fair assessment we present a summary of the current rebuttal status below:

| Reviewer   | Core Concerns                                                                                                                               | Our Targeted Rebuttal Work                                                                                                                                | Subsequent Feedback                                                                                                                               |
|:-----------|:----------------------------------------------------------------------------------------------------------------------------------------------|:------------------------------------------------------------------------------------------------------------------------------------------------------------|:--------------------------------------------------------------------------------------------------------------------------------------------------|
| Reviewer tCfY | Clarified the theoretical contribution and discuss why reverse-order pruning should be a baseline control for future pruning papers.                | We define our theoretical contribution as "a successful application of previous analyses" and explain why our method can serve as a general baseline from the perspectives of simplicity, effectiveness, and theoretical basis. | -                                                                                                                                                 |
| Reviewer mE3b |  Questioned the performance on mathematical tasks, perplexity, mixture of experts models.                                                         | Additional experiments demonstrate the effectiveness of our method on the GSM8K dataset and Mixtral 8×7B model, while also exhibiting lower perplexity.      | Time: 26 Nov 2025, 16:58 Comment: “Thank you for you answer, it answers my questions and **I've revised my score** accordingly.”             |
| Reviewer Ncfm |  Questioned the performance on mathematical and code-related tasks and experiments of reverse-order pruning with fine-tuning of the last few layers. | Additional experiments demonstrate that our method can achieve superior performance in mathematical and code benchmarks and reverse-order pruning with fine-tuning of the last few layers outperforms the cosine similarity-based approach. | Time: 25 Nov 2025, 19:54 Comment: “Thank you for the additional experiments. The results align with my expectations… **I have raised my score to 8 to support the paper's acceptance**.” |
| Reviewer PD7z |  Questioned experiments on width vs layer pruning and LoRA vs. partial fine-tuning experiments.                                                  | Additional experiments demonstrate that our method has a larger throughput than channel pruning, and can achieve almost the same performance as training a model from scratch with a much lower computational cost than training a model of the same size from scratch. | Time: 24 Nov 2025, 22:10 Comment: “In light of this and previous experimental results **I am willing to revise my assessment of the paper**.”   |

We respectfully request the AC to consider these substantial improvements and the rigorous validation provided.

---

### Meta-Review · Area_Chair_L6Go · 2026-01-06

**Summary:**

This paper offers a comprehensive reassessment of layer pruning for Large Language Models (LLMs), introducing a straightforward "reverse-order" pruning strategy (removing deep layers first) combined with partial fine-tuning of the final layers. The research demonstrates that this simple method outperforms existing complex pruning metrics and provides theoretical support based on gradient flow in Pre-LN architectures.

This work establishes a robust and practical baseline for the field. In the rebuttal phase, the authors performed additional critical experiments, including mathematical and code reasoning tasks (GSM8K, HumanEval), MoE architecture validation (Mixtral), and throughput comparisons. These additions effectively resolved reviewers' concerns regarding generality and efficiency, resulting in a consensus for acceptance.

**Reviewer Concerns:**

The majority of reviewers (3 out of 4) increased their scores after the authors' rebuttal (except for Reviewer tCfY, who initially gave a high score of 8).

**Addressed review**:

1. **Generalization to Reasoning/Generation Tasks (Reviewer mE3b, Ncfm)**: Reviewers initially questioned the reliance on multiple-choice benchmarks. The authors added experiments on GSM8K (math), HumanEval (code), and WikiText-2 (perplexity), showing the method outperforms baselines like ShortGPT and SliceGPT on these generation-heavy tasks.
2. **Applicability to Different Architectures (Reviewer mE3b)**: Concerns about applicability to Mixture-of-Experts (MoE) were addressed by new experiments on Mixtral 8x7B, where the method outperformed the UMD baseline.
3. **Efficiency vs. Width Pruning/Quantization (Reviewer PD7z, tCfY)**: Reviewers questioned the practical value of layer pruning compared to width pruning or quantization. The authors provided throughput benchmarks showing their method ($30.92$ tokens/s) is faster than width pruning ($25.76$ tokens/s) and demonstrated that layer pruning is orthogonal to quantization (compatible with GPTQ).
4. **Fine-tuning Strategy Robustness (Reviewer PD7z)**: Questions regarding the optimality of fine-tuning only the "last 3 layers" were addressed with ablation studies showing diminishing returns when tuning more layers, validating the efficiency of their proposed recipe.

**Reviewer Scores:**

- Reviewer tCfY: 8.

- Reviewer Ncfm: 6 -> 8.

- Reviewer mE3b: 4 -> 6.

- Reviewer PD7z: 2 -> 4.

---

### Decision · Program_Chairs · 2026-01-26

Accept (Poster)